# ONLINE LEARNING OF MULTIDIMENSIONAL DISTRIBUTIONAL MAPS FOR RAPID POLICY ADAPTATION

## ABSTRACT

In order to flexibly behave in dynamic environments, agents must learn the temporal structure of causal events. Standard value-based approaches in reinforcement learning (RL) learn estimates of temporally discounted average future reward, leading to ambiguity about future reward timing and magnitude. Recently, midbrain dopamine neurons (DANs) have been shown to resolve this ambiguity by representing distributional predictive maps of future reward over both time and magnitude in the encoding of reward prediction errors. However, the computational function of such time-magnitude distributions (TMD) in the brain is unknown. Here we present distributional neural learning (DNL), online learning rules for acquiring information-maximising multidimensional distributional estimates, extending classic work in distributional RL from 1D return distributions to efficient representations of distributions of arbitrary dimensionality. In previous distributional RL approaches, the distributional information is largely used for improving representation learning. In our framework, TMDs are the direct substrates for simple policy decoders, enabling rapid risk-sensitive action selection in environments with rich probabilistic temporal reward structure, even under distributional shifts. Finally, we present cross-species neural and behavioral evidence, from rodents and humans, consistent with the implementation of this theory in biological circuits. Our results advance a principled computational link between distributional RL and neural coding theory, and establish a role for multi-dimensional distributional predictive maps in rapidly generating sophisticated risk-sensitive policies in environments with complex, multi-modal, distributions of future reward.

## 1 INTRODUCTION

One of the most fruitful intersections between natural and artificial intelligence research has been the idea that midbrain dopamine neurons (DANs) in the brain encode a reward prediction error critical for reinforcement learning (RL) (Schultz et al., 1997), a suite of algorithms enabling an agent to learn to select actions based on reward feedback (Sutton & Barto, 2018). Recently, these neurons have been found to represent a diverse set of state features (Lee et al., 2024) including the distributional coding of the timing and magnitude of reward (Sousa et al., 2025). Beyond previous influential models of distributional magnitude coding (Dabney et al., 2020), these recent empirical results (Sousa et al., 2025) extend the 1D value code to a 2D time-magnitude distribution (TMD) of future reward. The novel identification of these TMDs within dopaminergic circuits raises the following critical questions. **Q1**. how are these representations acquired? **Q2**. how are they used for action selection?

> We aim to address this question by developing a comprehensive theory of such distributional predictive maps across biological and artificial agents based on the core idea that TMDs greatly simplify the problem of risk-sensitive sequential action selection. In particular, we focus on naturalistic settings where agents must choose between actions yielding probabilistic rewards generated by environments with complex temporal structure and shifting distributions.

Significant advances in state-of-the-art RL have been achieved through distributional RL algorithms which learn a one-dimensional probability distribution of value, rather than a single scalar sufficient statistic of value (e.g. the mean) in an online manner via environmental feedback (Dabney et al., 2018b; Rowland et al., 2019). In Section 2.2 we present a novel generalization of the one-dimensional

distributional learning rules to arbitrary dimensionalities thus providing *a mechanism by which multi-dimensional TMDs may be learned* **[Q1]**. This is accomplished by combining principles of distributional RL with efficient coding drawn from theoretical neuroscience (Ganguli & Simoncelli, 2014). In neuroscience, the theory of efficient coding models how sets or populations of neurons to maximize the amount of information they encode about diverse stimuli in the environment Barlow (1961); Olshausen & Field (1996); Yerxa et al. (2020). However it lacks a key element in brain computation: how neural population responses are learned. We address this by proposing a theory of efficient learning that generalizes distributional RL rules through optimal transport (Santambrogio, 2015).

Prior work in distributional RL has primarily focused on learning return distributions, leading to improved representation learning in deep distributional agents, but used scalar value estimates to generate policies (Bellemare et al., 2017; Dabney et al., 2018b). Less attention has been given to how these representations can support flexible risk-sensitive policies across time (Gagne & Dayan, 2021). Anticipating the timing of future rewards is especially important for sequential decisions such as foraging, where animals must balance the energetic costs of search with uncertain resource availability, and similarly, agents in RL must plan across extended horizons, trade off immediate versus delayed gains, and adapt their strategies to dynamic environments. In section 5 we show how having access to TMD representations allows for efficient solutions to sequential decision-making tasks with complex temporal dynamics. In section 6, we show how having access to TMD representations allows for *generating complex risk-sensitive behavior for arbitrary joint reward distributions in magnitude and time using simple linear readouts* **[Q2]**.

RELATED WORK

Distributed neural codes have been proposed to represent reward timing (Tiganj et al., 2019; Brunec & Momennejad, 2022; Masset et al., 2025), as well as the joint representation of reward magnitude and timing (Tano et al., 2020), future state occupancy (Brunec & Momennejad, 2022), and other task relevant features (Lee et al., 2024). However, these coding schemes are not efficient in the information-theoretic sense consistent with the long-standing hypothesis established in sensory neuroscience (Barlow, 1961; Ganguli & Simoncelli, 2014), they do not maximize information about reward under constraints on the population size and adapt to environment stimuli distributions. This is essential as the number of represented feature distributions becomes large. Furthermore, new data shows that when the reward TMD changes, the population of DANs adapt to encode the new distribution while preserving the relative tuning across the population (Sousa et al., 2025; Rothenhoefer et al., 2021). Relatedly, this is also observed for hippocampus and medial entorhinal cortex population codes in spatial navigation paradigms (Krupic et al., 2018; Boccara et al., 2019). More recently, efficient distributional RL models for the encoding of reward magnitude in midbrain DANs have been introduced (Schütt et al., 2024; Dabney et al., 2020), supported by mechanistic models and experimental evidence suggesting that direct and indirect striatal medium spiny neurons may implement such strategies (Lowet et al., 2025). However, these models are restricted to one-dimensional reward features or assume the dimensions are statistically independent (Sousa et al., 2025).

In machine learning, mean maximum mean discrepancy (MMD) and regularized Wasserstein losses have been applied to learn multidimensional distributions (Sun et al., 2024; Wiltzer et al., 2024; Zhang et al., 2021). However, these approaches regularize the optimized distribution, whereas DNL regularizes the learning trajectory. In dimensions bigger than one, this distinction is critical: there are many distinct particle configurations that represent the same distribution, which leads to degenerate solutions (Fig. 1). We address this degeneracy by introducing a Wasserstein regularizer on the smoothness of the learning trajectory, enforcing the conservation of relative particle positions throughout learning. This modification yields three key benefits: (1) it extends efficient coding principles to higher dimensions (Schütt et al., 2024); (2) it preserves relative particle arrangements as the reward distribution changes, matching empirical observations across multiple brain areas and modalities; and (3) it supports flexible decoding of subjective value (i.e., risk-sensitive utility) under context-dependent time–magnitude distributions (TMDs), removing the need to retrain the decoder.

## 2  WASSERSTEIN GRADIENT FLOW FORMULATION FOR DISTRIBUTIONAL RL AND EFFICIENT CODING

A core hypothesis regarding brain function is the idea that populations of neurons efficiently encode environment features (Barlow, 1961). A long-standing limitation of efficient coding (EC) theory is that it does not provide algorithms for learning efficient representations of environmental statistics through online interactions, a key requirement for adaptive behavior (Ganguli & Simoncelli, 2014; Yerxa et al., 2020) (Fig. 1a). In contrast, distributional RL offers online update rules for constructing efficient representations (Schütt et al., 2024) of value distributions in the environment (Dabney et al., 2018b) (Fig. 1b), though these methods are typically restricted to one-dimensional distributions. We begin by showing that both EC and distributional RL can be expressed as optimal transport (OT) problems that minimize the Wasserstein distance between the representations and environmental statistics (Santambrogio, 2015).

Both EC and quantile regression distributional RL algorithms (Dabney et al., 2018b) approximate a target distribution $p(r)$ of an environment feature $r$, such as reward, non-parametrically using a finite set of units $\{\theta_i\}_{i=1}^N$, with $\theta_i \in \mathbb{R}$, which define an approximate distribution

$$q(r) \approx \frac{1}{N} \sum_{i=1}^N \delta_{\theta_i}(r) \,, \tag{1}$$

where $\delta$ is the Dirac delta function (Wiltzer et al., 2024; Zhang et al., 2021). In EC, the goal is to optimize $\theta_i \in \mathbb{R}$ in order to maximize the mutual information between neural population responses, which imply an approximate posterior $q_\lambda(r)$, and the target $p(r)$, while constraining the number of units $N$ (Sousa et al., 2025). In quantile regression distributional RL, the goal is to predict future rewards, therefore units $\theta_i$ are iteratively updated given reward samples $r'$, to minimize the Wasserstein distance between the approximate distribution $q(r)$ and the target $r' \sim p(r)$, eventually converging to the quantiles of the target distribution (Dabney et al., 2018b).

The Wasserstein distance is defined in terms of the transport map which describes how to reshape the probability mass of $q(r)$ to match $p(r)$ (Fig. 1c). Let $\mathcal{T} : \mathbb{R} \to \mathbb{R}$ denote the set of transport maps from an approximate distribution $q(r)$ to a target $p(r)$. The 2-Wasserstein distance is defined as

$$W_2(q, p) = \left( \inf_{T \in \mathcal{T}} \int |r' - T(r')|^2 dp(r') \right)^{1/2} \,, \tag{2}$$

and intuitively measures the minimum transport cost, measured as the total Euclidean distance, required to reshape one distribution into another. In the one-dimensional case only, the OT $T^*$ has a closed-form solution given by mapping the quantiles of $q(r)$ to $p(r)$:

$$W_2(q, p) = \left( \int_0^1 |Q^{-1}(u) - P^{-1}(u)|^2 du \right)^{1/2} \,, \tag{3}$$

where $Q$ and $P$ are the cumulative distribution functions (CDFs) of $q$ and $p$ respectively. Setting $q$ to be the uniform distribution, leads to the one-dimensional efficient code proposed by Ganguli & Simoncelli (2014), where units are transported to the target distribution $p(r)$ through the the inverse cumulative distribution function $T^* : u \to P^{-1}(u)$ (Fig. 1a). Specifically, mapping through the inverse CDF maximizes the mutual information $I(\lambda, s)$ between the target variable $r$ and the neural population response $\lambda$ (Fig. 1a, (Ganguli & Simoncelli, 2014)). Therefore, both EC and distributional RL correspond to the problem of identifying a Wasserstein-optimal transport map between the non-parametric approximate distribution $q(r) \approx \frac{1}{N} \sum_{i=1}^N \delta_{\theta_i}(r)$ and a target distribution $p(r)$, and Eqn. 3 provides a unique solution in the one-dimensional case.

In the next section, we introduce a Wasserstein gradient flow formulation that generalizes this framework to arbitrary dimensions and show that distributional RL arises as a stochastic approximation of it (Santambrogio, 2015). This forms the theoretic basis for our development of online multi-dimensional learning rules for multi-dimensional target distributions which unifies EC in neuroscience with distributional RL.

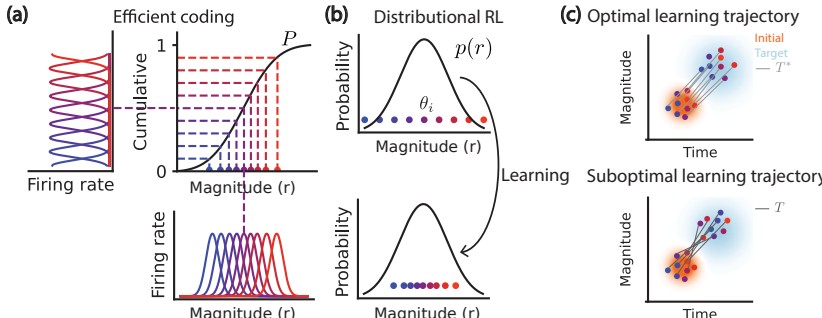

Figure 1: (a) Uniformly distributed units with Gaussian tuning functions are mapped to an efficient one-dimensional code using the inverse CDF. (b) The top plot shows an initial set of uniformly distributed units, which, through online updates driven by reward samples, converge to the quantiles of the reward distribution shown below. (c) The top plot illustrates units representing the initial distribution (orange) being transported to the target distribution (blue) along the optimal learning trajectory, while the bottom plot shows a suboptimal trajectory. In both cases, the final units approximate the target distribution equally well, but the top trajectory achieves faster convergence.

## 2.1 FROM WASSERTEIN DISTANCE IN 1-D TO WASSERSTEIN GRADIENT FLOW IN N-D

For one-dimensional feature spaces, the inverse CDF defines a unique solution (Eqn. 3). The critical obstruction to generalizing to higher dimensions is that there are an infinite number of configurations of particles which can produce the same approximate distribution (Fig. 1c). To resolve this degeneracy, we additionally regularize the smoothness of the learning trajectory toward the target distribution. This allows the preservation of the relative position of the particles, which is crucial for leveraging these distributions in downstream tasks.

In order to do this, we consider the evolution of an initial approximate density $q(r)$ to a target density $p(r)$ under a potential $\Phi(r)$ and a diffusion constant $\beta^{-1}$ which can be described by the Fokker–Planck equation (FPE) $\frac{\partial q(r,t)}{\partial t} = \nabla_r \cdot (q(r,t)\nabla_r\Phi(r)) + \beta^{-1}\Delta_r q(r,t)$, that converges to an equilibrium distribution $p(r) \propto e^{-\beta\Phi(r)}$. Setting $\Phi(r) = -\frac{1}{\beta}\log p(r)$ ensures convergence to the desired $p(r)$ corresponding to an environment feature distribution in EC or a target reward distribution in the simplified distributional RL setting we are considering for exposition. This induces a probability flow on $q(r)$ toward high-probability regions to match $p(r)$, while also inducing an inductive bias toward *smooth* learning trajectories. However, this process is offline, as it requires full knowledge of $p(r)$ in advance.

A classical result approximates the solution to the FPE by Wasserstein gradient descent on a variational free energy functional, e.g. the Kullback-Leibler (KL) divergence, in the space of distributions (Jordan et al., 1998; Otto, 2001)

$$q_{t+1} = \arg\min_{q'}\{D_{\mathrm{KL}}\left[q'||p\right] + \frac{1}{h}W_2^2(q', q_t)\}, \tag{4}$$

where $h$ is the update step size. Crucially, this optimization can be carried online, by stochastically approximating $p$ using samples. Motivated by this connection, we adopt a particle-based approximation as defined in Eqn. 1 that jointly minimizes the KL divergence to the target and the Wasserstein distance between successive approximate distributions (Chen et al., 2018). This formulation generalizes EC to multiple dimensions, preserving the relative positioning of particles through learning, and resolves the degeneracy inherent for high-dimensional representations. While previous multidimensional distributional RL proposals only regularize to the target distribution (Sun et al., 2024; Zhang et al., 2021; Wiltzer et al., 2024) (i.e. $W_2^2(q', p)$), we regularize the particle trajectory ($W_2^2(q', q_t)$).

For clarity of exposition, we focus on the TMD, the joint distribution of reward times ($t$) and magnitudes ($m$): $p(t, m)$. However, our theory generalizes to other stimulus dimensions, such as spatial occupancy and reward in space, which we explore in simulation in order to explain empirical neural recordings from the brain.

## 2.2 LEARNING MULTIDIMENSIONAL EFFICIENT POPULATION CODES

In multidimensions, we optimize Eqn. 4 considering approximate TMD $q_t \in \mathbb{R}^2$ is represented by a set of units $\{\boldsymbol{\theta}_i\}_{i=1}^N$, $q_t(\mathbf{r}) = \frac{1}{N}\sum_i \delta_{\boldsymbol{\theta}_i}(\mathbf{r})$. Each unit $\boldsymbol{\theta}_i = (t_i, r_i) \in \mathbb{R}^2$ encodes a reward delay $t_i$ and magnitude $r_i$. We decompose Eqn. 4 into two forces: one that attracts the units towards the reward samples ($F_1$) and a second one that regularizes the unit interactions, such that they do not occupy the same position ($F_2$)

$$q_{t+1} = \arg\min_{q' \in \mathcal{Q}} \{ \underbrace{-\mathbb{E}_q[\log l(\boldsymbol{\theta}|\boldsymbol{r})]}_{F_1} + \underbrace{\mathbb{E}_{q'}[\log q'] + \frac{1}{h}W_2^2(q', q_t)}_{F_2} \} \ , \tag{5}$$

where $l(\boldsymbol{\theta}|\boldsymbol{r})$ is the likelihood function of unit parameters given observed reward samples (Chen et al., 2018). Importantly, this discrete gradient flow is guaranteed to converge in the large sample limit Jordan et al. (1998). Introducing an entropic regularization with weight $\lambda$ on the joint distribution over particle pairs across iterations leads to the the DNL update:

$$\boldsymbol{\theta}_i^{t+1} \leftarrow \boldsymbol{\theta}_i^t - \alpha \left( \frac{\partial F_1}{\partial \boldsymbol{\theta}_i} + \frac{\partial F_2}{\partial \boldsymbol{\theta}_i} \right) \ ,$$

where

$$\frac{\partial F_1}{\partial \boldsymbol{\theta}_i} = -\nabla_{\boldsymbol{\theta}_i} \log l(\boldsymbol{\theta}_i|\boldsymbol{r}) \quad \text{and} \quad \frac{\partial F_2}{\partial \boldsymbol{\theta}_i} = \sum_j c \left( \frac{d_{ij}}{\lambda} - 1 \right) e^{\frac{-d_{ij}}{\lambda}} (\boldsymbol{\theta}_i - \boldsymbol{\theta}_j^t) \ , \tag{6}$$

$d_{ij} = \|\boldsymbol{\theta}_i - \boldsymbol{\theta}_j^t\|^2$, $c$ controls the strength of pairwise regularization, and $\lambda$ determines the interaction range between units and incorporates the step size $h$. When $\frac{d_{ij}}{\lambda} > 1$, $\boldsymbol{\theta}_i$ is attracted toward $\boldsymbol{\theta}_j$ with a force proportional to $(\frac{d_{ij}}{\lambda} - 1)e^{-d_{ij}/\lambda}$. Conversely, when $\frac{d_{ij}}{\lambda} < 1$, the interaction is repulsive. Increasing $\lambda$ makes the interactions progressively more global, such that all units influence one another as $\lambda \to \infty$. A full derivation of Eqns. 5 and 6 is provided in Supplementary Section A.3. For numerical simulations, we estimate $l(\boldsymbol{\theta}|\mathbf{r})$ by smoothing reward samples with a radial basis function kernel and evaluating the resulting density at each $\boldsymbol{\theta}_i$.

## 2.3 APPROXIMATELY RECOVERING THE 1-DIMENSIONAL DISTRIBUTIONAL RL

We demonstrate that the DNL multidimensional learning algorithm is an extension of the one-dimensional expectile learning rule Dabney et al. (2018b). In one dimension, we consider the $\lambda \to \infty$ limit, where all units contribute to the $F_2$ update of all other units. In this limit, the units can be ordered from lower to higher values, which allows averaged interactions with a given unit $\theta_i$ to be approximated by: 1) $F_2$ term of the mean of units to the left ($\bar{\theta}_{\text{LEFT}}$) scaled by the number of units on the left ($\tau$); and 2) $F_2$ term with the mean of the units to right ($\bar{\theta}_{\text{RIGHT}}$) scaled by the number of units of the right ($1 - \tau$)

$$\frac{\partial F_2}{\partial \theta_i} \approx \tau c(\bar{\theta}_{\text{LEFT}} - \theta_i) + (1 - \tau)c(\bar{\theta}_{\text{RIGHT}} - \theta_i) \ ,$$

where $r$ is a reward sample. Additionally, assuming a Gaussian likelihood $l(\theta_i|r) \sim \mathcal{N}(r, \beta^2)$, our learning rules become

$$\frac{\partial F_1}{\partial \theta_i} = -\frac{1}{\beta^2}(r - \theta_i) \quad \text{and} \quad \frac{\partial F_2}{\partial \theta_i} \approx \tau c(\bar{\theta}_{\text{LEFT}} - \theta_i) + (1 - \tau)c(\bar{\theta}_{\text{RIGHT}} - \theta_i) \ .$$

We notice that $\bar{\theta}_{\text{LEFT}} \approx \mathbb{E}[r|r \leq \theta_i]$ and $\bar{\theta}_{\text{RIGHT}} \approx \mathbb{E}[r|r > \theta_i]$, hence considering a sample $r$ and additionally setting $\beta = c = 1$ we recover the expectile learning rules (Fig. 2a):

$$\theta_i \leftarrow \theta_i + \tau \delta_{r > \theta_i}(r - \theta_i) + (1 - \tau)\delta_{r < \theta_i}(r - \theta_i).$$

A full derivation of the previous equations is in Section A.4 of the SM.

## 3 COMPARING DISTRIBUTIONAL NEURAL LEARNING WITH RELATED WORK

For non-factorizable joint distributions over two or more dependent random variables, DNL and MMD accurately represents the full joint distribution, whereas the factorized one-dimensional

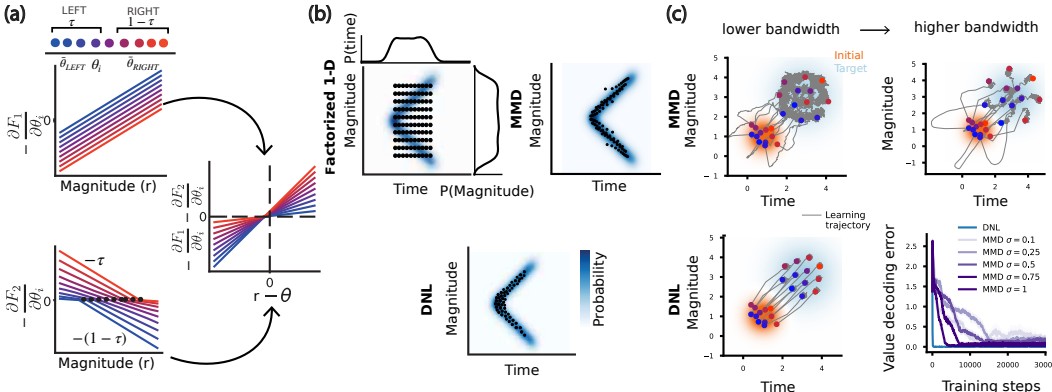

Figure 2: (a) Illustration of the DNL rules for 1-dimensional reward magnitudes. Units are color coded by the relative positioning toward $r$. (b) Samples were generated from the v-shaped reward TMDs represented in blue. The converged units for the factorized one-dimensional distributional learning, the MMD, and the DNL are represented as black dots. (c) Units learn to go from the initial to target distribution for MMD (top) and DNL (bottom). Learning trajectories are represented in grey. Bottom right: Value error decoding (considering 25% of units closest to origin) as a function of training steps for DNL and MMD (considering Gaussian kernels with different bandwidths).

quantile learning fails (Fig. 2b). Furthermore, in the SM Section A.6 we include simulations that investigate the inductive biases introduced by deep DNL and compare with the previously proposed 1-dimensional Deep QR-RL. We stacked DNL across network layers and trained the model end-to-end with backpropagation. Crucially, OT regularization in DNL biases learning toward smooth, connected solutions, improving generalization over deep QR-RL.

Additionally, OT regularization enforces the preservation of relative positioning of units along the learning trajectory (Fig. 2c). This property enables risk-sensitive value decoding to transfer across distribution shifts without retraining, unlike MMD-based approaches Zhang et al. (2021). In Fig. 2c, we simulate a population of units transported to a target distribution. Compared to MMD, DNL learning trajectories are more *direct* and preserve population geometry. Therefore, considering a risk-sensitive linear value decoder, DNL transfers more efficiently to the target distribution than MMD (Fig. 2c, top).

## 4 LEARNING MULTIDIMENSIONAL DISTRIBUTIONS IN THE BRAIN

We apply our DNL theory to the adaptation of midbrain dopamine in a classical conditioning experiment with mice where different cues predict reward at distinct delays and reward distribution (Sousa et al., 2025). In this study, DANs were shown to encode the probability of rewards over time and magnitudes (Sousa et al., 2025) (Fig. 3a). Additionally, the probability distribution of reward times was manipulated by removing either the cue that predicts the shortest or longest delay (Fig. 3c) and the probability distributions over magnitudes was also manipulated by providing separate cues predicting variable or certain magnitudes. The tuning functions of midbrain DANs were observed to adapt to the new distributions in time and magnitude (Fig. 3c).

We modeled each DAN tuning function based on the preferred reward time and magnitude $\boldsymbol{\theta}_i = (t_i, r_i)$. Importantly, as DANs exhibit tuning functions in time characterized by exponentially decaying temporal discount factors ($\gamma$), the tuning towards reward times $t_i$ was mapped into $\gamma_i$ considering: $\gamma_i = e^{-\frac{1}{t_i}}$. We simulated the tuning function adaption of DNL learning rules using simulated observations from the distributions of reward times and magnitudes given in the experiment Sousa et al. (2025). In agreement with experimental observations, DNL predicts that temporal discount factors became steeper when shifting to shorter reward times compared to longer ones (Fig. 3c). Conversely, sensitivity to reward magnitudes was more variable for cues predicting uncertain reward than for those predicting fixed rewards (Fig. 3c). In the SM Fig. 10 illustrates adaptation of hippocampal place cells and entorhinal cells to spatial structure and reward location changes.

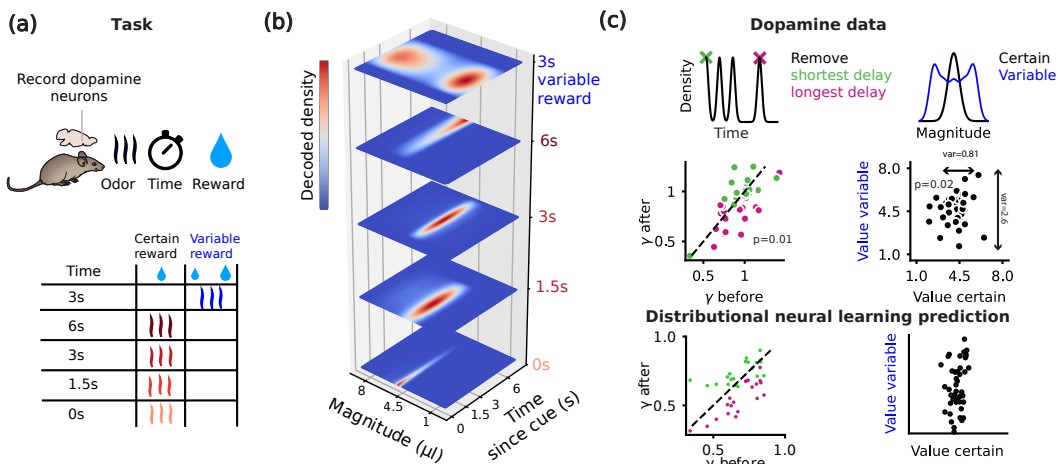

Figure 3: DNL models firing of DANs. (a) The task consists of four cues predicting a certain reward magnitude after a variable delay and an additional cue predicting a variable reward after a 3s delay. (b) The decoded joint density of reward over magnitude and time, using the DAN responses aligned to the different cues. Adapted from (Sousa et al., 2025) (c) Left: Adaptation of midbrain DANs' temporal discount factors when shifting to shorter or longer reward times by removing the shortest (magenta) or longest (green) reward delay. Right: Adaptation of DANs tuning for cues predicting variable (blue) and certain (black) reward magnitudes. Bottom: predictions from DNL theory.

## 5 REWARD TIME AND MAGNITUDE DISTRIBUTION FOR GENERALIZABLE RL

Learning flexible policies is essential for animals navigating environments with probabilistic rewards. For example, an animal may observe clouds and infer a probability of rain at a later time or may hear a sound and infer the location and timing of moving prey. To make informed decisions over time, animals must integrate learned associations between stimuli and probabilistic reward magnitudes and delays. Yet, prior applications of distributional RL in both neuroscience and machine learning have been limited to learning only one-dimensional reward distributions, primarily to support flexible risk sensitivity Dabney et al. (2018a); Ávila Pires et al. (2025). On the other hand, the multidimensional distributional methods described above Wiltzer et al. (2024); Zhang et al. (2021) have focused on modeling distributions over multiple sources of reward, rather than over multiple attributes (such as time and magnitude) of a single reward source. Here we demonstrate the advantages of learning reward time-magnitude distributions, similar to those decoded from the activity of DANs (Sousa et al., 2025), for reinforcement learning tasks.

First, we consider a **patch environment** in which several stimuli each indicate a possible reward at a given patch after a delay. In this simple environment, each patch is connected to all others as shown in Fig. 4a, and action $a_t = j$ deterministically brings the agent to patch $j$ at time $t + 1$. At every time step, regardless of the current state of the agent, the probability of stimuli $i$ appearing was drawn with constant probability, independent of all other stimuli. Multiple stimuli were allowed to appear at the same timestep and a second stimulus could occur before the delivery of reward from a previous stimulus. Each stimulus was associated with a probabilistic reward TMD such that a reward was drawn from a distribution of possible reward magnitudes and delays after the stimulus onset. The reward was available only at this delay and disappeared afterward, regardless of consumption. We compare a time-magnitude reinforcement learning (TMRL) agent which learns a reward TMD for each stimulus with an agent using a fully enumerated state space with a unique state for every possible combination of time passed since each stimulus.

To predict the value of each state at the next time step conditioned on past stimuli, the TMRL agent only needs to keep track of the elapsed time since each of the stimuli and use a learned TMD for each stimulus, specifically $\text{TMD}(s, t, r) \approx \frac{1}{N} \sum_i \delta_{\boldsymbol{\theta}_i^s}(r, t)$ using units $\boldsymbol{\theta_i} = (\tau_{\boldsymbol{\theta_i}}, r_{\boldsymbol{\theta_i}}) \in \mathbb{R}^2$ learned through the DNL update rules in Eq. 6. A full description of how the TMRL selects actions is included in the SM Algorithm 1. The standard RL agent learns a Q-value through temporal difference (TD) learning ($Q(s, a) \leftarrow Q(s, a) + \alpha(r(t) + \gamma \max_{a'} Q(s', a') - Q(s, a))$, where the state $s$ indicates

current location and the time elapsed since each stimuli was observed, the action $a$ is a movement direction, $\alpha$ is the learning rate and $r(t)$ is the reward encountered at time t. Fig. 4b shows the TMRL agent is able to learn the TMD quickly and adapt these distributions to various combinations of stimulus times. On the other hand, the standard RL agent must experience and learn action values for each possible stimulus delay combination and therefore, requires more samples from the environment to optimize. The state space of the standard agent grows exponentially with the number of states and stimuli and we demonstrate the effect this has upon the learning rate (Fig. 4c).

To validate that this performance improvement is due to the multidimensionality of the time-magnitude distributions, in Supp Fig. 11 we include agents with the same learning as TMRL but with the time or magnitude dimension ablated. The asymptotic reward rate achieved by these agents using only the expected value in either the time or magnitude dimension is inferior to the full TMRL agent.

Next, we consider a similar environment, again with several stimuli indicating probabilistic rewards after a delay, but in a **gridworld** as represented in Fig. 4d. The standard RL agent once again must fully enumerate the state space with both time from each stimulus and location on the grid. However, the TMRL agent now learns multiple successor representations (SR) (Dayan, 1993) expanded in delay time as well as a TMD for each of the stimuli, as depicted in Supp. Fig. 14. These SRs $\hat{M}_i^{\pi_i}(s, s', t')$ learn separate policies $\pi_i$ that bring the agent to different reward locations and allow flexible combinations of these policies using generalized policy improvement (GPI) when multiple rewards are available with various time delays. The SR is the state occupancy of state $s'$ starting in state $s$ following policy $\pi$ for $t$ time steps (Dayan, 1993) (with maximal time steps $\tau_D$). By combining these SRs with the current probabilistic reward time-magnitude distribution $\text{TMD}(s', t', r)$ determined by the timings of recent stimuli, the agent can use GPI to select actions (Barreto et al., 2016):

$$V(s) = \sum_{s', t' \in \{1,..,t_D\}, r'} r' \, M(s, s', t') \text{TMD}(s', t', r') \, . \tag{7}$$

Fig. 4e shows the TMRL agent outperforms the standard RL and the QR-RL agent and that this difference grows as the number of stimuli increase, demonstrating the utility of flexibly combining reward time-magnitude distributions in sequential decision making. Importantly, the QR-RL agent is learning a distribution over value using the same state space as the standard RL agent. Implementation details of the agents as well as the environment are left to the SM Section B.3.2.

# 6 Decoding policies in multidimensional risk-sensitive RL

In this section we investigate how the TMD representations found in DANs may support risk-sensitive behavior in reward time and magnitude. For 1-dimensional reward magnitude distributions, risk sensitivity behavior can be generated by assigning weights to the reward distribution quantiles (Dabney et al., 2018a). For example, overweighting lower reward quantiles generates risk-averse behavior, while overweighting higher quantiles instead produces risk-prone behavior. We extend risk sensitivity to reward time and magnitude by applying weights that depend on both reward magnitudes and delays to compute the subjective value: $V(s) = \sum_{s', t' \in \{1,..,t_D\}, r'} r' \, M(s, s', t') w(t', r') \text{TMD}(s', t', r')$. To model risk sensitivity in sequential RL tasks, we consider the gridworld environment in Fig. 4d with different stimuli structure and weighting functions $w : \mathbb{R}^2 \to \mathbb{R}$, such that $w(t, r)$ is the weight assigned to the reward time $t$ and magnitude $r$.

**Magnitude risk**   To demonstrate risk sensitivity under reward uncertainty, we simulate the gridworld with three stimuli: one predicting certain reward (certain); another predicting uncertain reward magnitudes with a higher expected value (risky); a control certain reward (control). During testing, the risky and certain stimuli are always presented simultaneously, forcing the agent to select between risky or certain options, while the control stimulus was presented randomly. The risk-sensitive agent computed a weighting $w(t, m)$ which only gave weights to smaller reward magnitudes. Figure 5b shows the TMRL agent often chose the risky choice with higher expected value but the risk-sensitive TMRL agent almost never chose the risky choice, opting for the certain option.

**Time varying risk**   Risk-sensitive behavior can also arise through internal state dynamics, and survival may depend on fast adaptation to these states. In Figure 5a top, the dynamics of an internal state such as satiety is shown over time (decaying exponentially between reward consumption). An

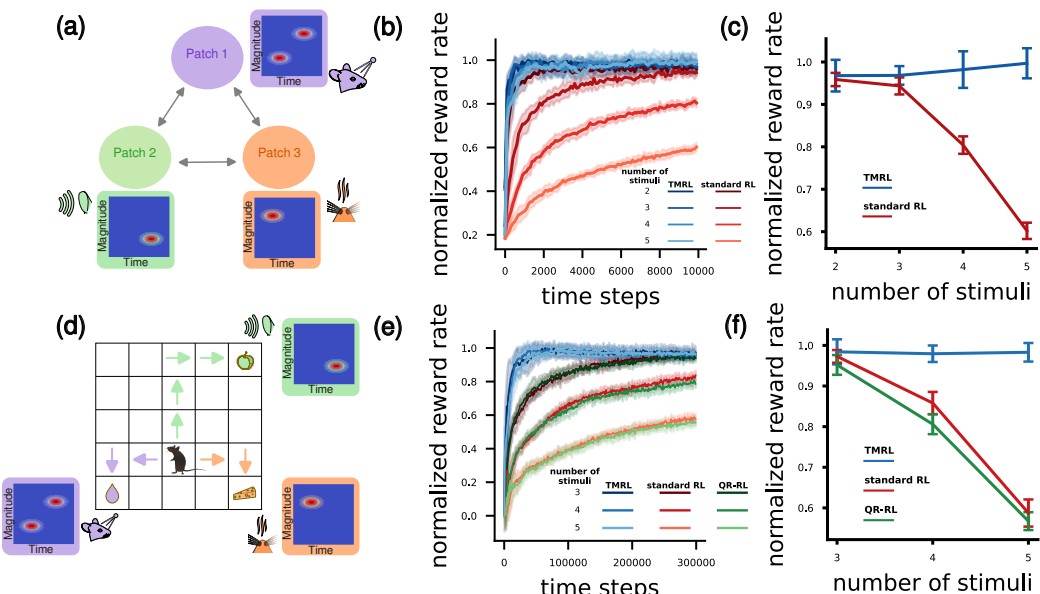

Figure 4: (a) Each stimulus indicates a reward given at the corresponding patch with a magnitude and delay after stimulus observation drawn from a TMD. (b) Reward rate of agents (TMRL: blue, standard RL: red) training over time steps on the patch environment. Reward rate is normalized to the largest mean reward achieved so that learning rate can be compared across environments with different number of stimuli. (c) Same data as b but only showing the final reward rate for the TMRL (blue) and standard RL (red) agents after 10000 training steps. (d) Gridworld environment in which a stimuli indicates reward at different locations with reward magnitude and delay drawn from different TMDs. (e) Reward rate of agents (QR-RL: green) training over time steps on the gridworld environment. Reward rate is normalized to the largest mean reward achieved. (f) Final reward rate for the TMRL (blue), standard RL (red), and QR-RL (green) agents after 300000 training steps.

agent may become risk averse with respect to the magnitude or timing of rewards during times of low satiety, requiring certain rewards as soon as possible. After satiation, when immediate rewards are not as critical, animals may revert to more risk prone behavior. To model this scenario, we consider two stimuli: one signaling a certain low reward; the other predicting a higher reward but occurring less frequently. Importantly, the expected value for the high-magnitude lower probability reward was larger, so an agent optimizing expected reward would favor it. A TMRL agent was compared to a time-varying risk-sensitive TMRL agent where weights $w(t, m|x)$ are a non-linear function of its satiety level $x_t$ (Supp. equation 41). The time-varying risk-sensitive TMRL agent achieves slightly lower expected reward rates (Supp. Fig. 16b) but a higher subjective value (i.e. the risk-modulated utility to the agent) as predicted (Fig. 5a).

Interestingly, human risky decision-making shows similar weighting schemes. To experimentally measure risk sensitivity of probabilistic rewards, human participants chose between certain and probabilistic rewards (Fig. 5c, Green et al. (1999a)). Fig. 5c plots the measured subjective value of a probabilistic option, defined as the reward magnitude of the certain option for which the participant is equally likely to choose the certain and probabilistic options. Similarly, to assess how individuals discounted delayed rewards, participants were asked to choose between immediate and delayed options in intertemporal experiments. In this case, the subjective value of the delayed reward was defined as the reward magnitude of the immediate option for which the participant was equally likely to choose the immediate and delayed options. Studies have shown that humans discount smaller rewards more steeply over time than larger ones, while they discount larger rewards more steeply under uncertainty (Green et al., 1999a). These behavioral asymmetries, seen in Fig. 5c as the swap in ordering of the dashed and full curves, have led to the hypothesis that distinct mechanisms underlie temporal and probabilistic discounting. We found that modeling risk-sensitive behaviors using a weighting scheme that decreases hyperbolically independently in delay and magnitude (**Factorized:** $w(t, r) = \frac{1}{(1+k_t t)^{s_t}} \cdot \frac{1}{(1+k_r r)^{s_r}}$) is insufficient to generate both behaviors (Fig. 5d).

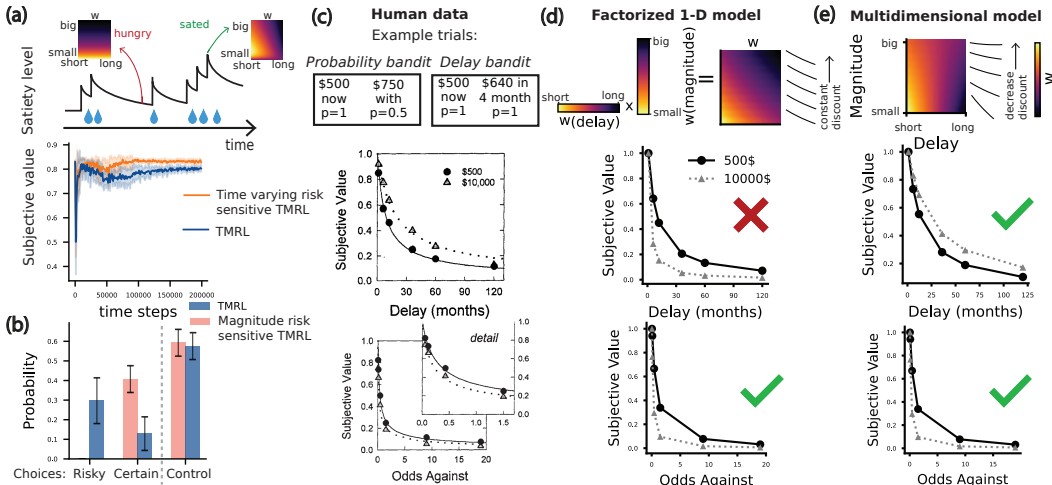

Figure 5: (a) Top: Satiety level example dynamics as a function of rewards. The weights $w_t$ applied to the TMDs depend on the satiety level at time t. Bottom: Subjective value for TMRL and time varying risk-sensitive agent. (b) Probability of choosing risky, certain and control option for the TMRL and magnitude risk-sensitive TMRL agent. (c) Top: example of a trial where participants choose between a certain and a risky option and a trial where participants choose between an immediate and delayed option. Humans discount larger rewards more steeply under uncertainty (middle) and smaller rewards more steeply over time (Green et al., 1999a). (d) Top: Factorized weights for reward magnitudes and times. Middle and bottom: Predicted subjective value for the model considering independent weights for reward time and magnitude. (e) Multidimensional weights where the delay discounting decreases with the magnitudes. Middle and bottom: Predicted subjective value that match human behavior.

However, we are able to reproduce the reverse discounting effect by instead considering weights that decrease hyperbolically jointly in delay and magnitude (**Multidimensional:** $w(t, r) = \frac{1}{(1+k(t+r))^s}$) (Fig. 5e)[1]. This multidimensional scheme reduces delay sensitivity for larger rewards, capturing the reverse discounting seen in humans (Fig. 5c,e). Such correlated weighting cannot be captured by independent dimensions, suggesting that risk sensitivity and intertemporal biases may stem from a shared mechanism Green et al. (1999a;b).

## 7 DISCUSSION AND LIMITATIONS

In this work, we unify the long-standing frameworks of efficient coding and distributional learning, proposing a neural learning model for the joint time–magnitude reward representations observed in midbrain dopaminergic neurons. Our theory specifies how population geometry should be iteratively reshaped throughout learning to optimize the learning trajectory. While we focused here on reward TMDs (Masset et al., 2025; Sousa et al., 2025), our DNL learning rules apply broadly to arbitrary stimulus dimensions and behaviorally relevant control variables (Hollup et al., 2001; Boccara et al., 2019; Ebitz & Hayden, 2021). However, our current formulation does not yet incorporate biological constraints such as metabolic cost, synaptic plasticity mechanisms, or recurrent feedback loops (Denève & Machens, 2016). Prior work in distributional RL typically derives policies from value estimates without fully exploiting the underlying reward distributions (Dabney et al., 2018b; Bellemare et al., 2017; Wiltzer et al., 2024; Zhang et al., 2021). To move beyond these approaches, we designed a naturalistic RL task with probabilistic variation in reward timing and magnitude, and show that agents equipped with predictive TMDs learn and generalize more rapidly in sequential decision-making tasks with complex temporal dynamics. While we have limited our experiments to tabular environments in order to precisely elucidate the functional role of TMDs, our algorithmic strategies may be incorporated into scalable architectures with deep function approximation. For example, TMDs may be learned over world model latents thus providing a pathway towards rapid risk-sensitive control in state-of-the-art benchmarks (Hafner et al., 2025).

---

[1]The $k$ parameter controls the discount rate and $s$ sets the shape of the discounting curve.

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

# A  APPENDIX

### REPRODUCIBILITY STATEMENT

We have included all source code in the supplementary material and have thoroughly verified the reproducibility of the experimental results. All proofs from derivations in the main text are included in the Appendix with clear statements of the underlying assumptions.

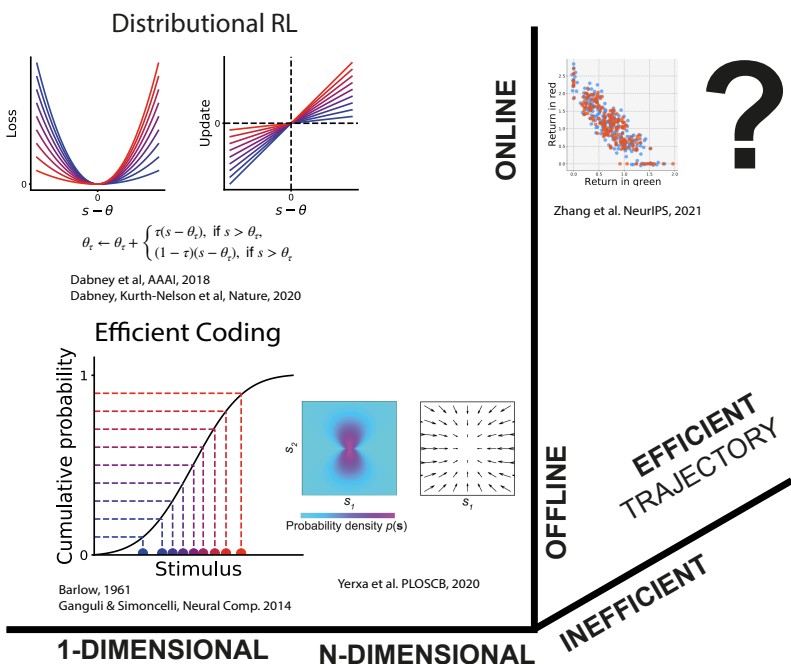

Figure 6: We summarize related work in terms of: 1) the dimensionality (from one-dimension to $n$-dimensional) of the stimulus space; 2) if the population code (quantile or neural) is learned through online interactions with the environment stimuli or optimized offline; 3) if the learning trajectory is efficient or not. Classical work (Barlow, 1961; Ganguli & Simoncelli, 2014) (one-dimensional stimuli) and recent generalizations (Yerxa et al., 2020) ($n$-dimensional stimuli) optimize population codes offline while distributional RL (Dabney et al., 2018b) provides learning rules for quantiles approximating one-dimensional distributions. Recently, multidimensional distributional RL approaches were proposed (Zhang et al., 2021; Sun et al., 2024; Wiltzer et al., 2024), however it leads to inefficient learning trajectories. Additionally, Generative adversarial networks (GAN) models have been also proposed to learn the distributions (Freirich et al., 2019; Doan et al., 2018). Our work fills in the blank upper-right quadrant with $n$-dimensional codes being learned through online efficient updating rules.

## A.1  DEFINING QUANTILES AND EXPECTILES

We briefly summarize the definitions of quantiles and expectiles of a distribution for reference. Essentially, quantiles generalize the median statistic of a distribution and expectiles generalize the mean statistic Rowland et al. (2019).

- The $\eta$-quantile $\theta$ of a random variable $R$ with probability mass function $p$ satisfies:

$$\eta p(r < \theta) = (1 - \eta)p(r \geq \theta).$$

  The median is the $0.5$-quantile ($\eta = 0.5$).
- The $\eta$-expectile $\bar{r}$ satisfies Newey & Powell (1987):

$$\eta \mathbb{E}[(\bar{r} - r)^-] = (1 - \eta)\mathbb{E}[(\bar{r} - r)^+] .$$

The expectiles are distributed according to the cumulative distribution,

$$P(\bar{r}) \sim \frac{\mathbb{E}[(\bar{r} - r)^+]}{\mathbb{E}[|\bar{r} - r|]} \quad .$$

The mean is the $0.5$-expectile ($\eta = 0.5$).

## A.2 COMPARING DISTRIBUTIONAL NEURAL LEARNING RULES WITH MAXIMUM MEAN DISCREPANCY LEARNING RULES

In (Zhang et al., 2021), a distributional learning algorithm is proposed for multi-dimensional reward functions based on the maximum mean discrepancy (MMD) (Gretton et al., 2012), extending the one-dimensional MMD-based algorithm in (Nguyen et al., 2020). The proposed learning algorithm, similarly to our algorithm, approximates the target distribution using a finite set of units. However, rather than minimizing the KL divergence, it minimizes the MMD between the approximated $q'$ and target $p^2$ distributions:

$$q_{t+1}(s) = \arg \min_{q' \in \mathcal{Q}} \{ \mathrm{MMD}^2(q'(s), p(s)) \} \quad , \tag{8}$$

where $\mathcal{Q} = \{ q' : q'(s) = \sum_{i=1}^{N} \frac{1}{N} \delta_{\theta_i}(s) \}$. Here we derive exact analytic update rules considering a Gaussian kernel $k$ with covariance matrix $\boldsymbol{\Sigma}$:

$$k(x, z) = \exp\left(-\frac{1}{2}(\boldsymbol{x} - \boldsymbol{z})^\top \boldsymbol{\Sigma}^{-1}(\boldsymbol{x} - \boldsymbol{z})\right), \qquad \nabla_{\boldsymbol{x}} k(\boldsymbol{x}, \boldsymbol{z}) = -\boldsymbol{\Sigma}^{-1}(\boldsymbol{x} - \boldsymbol{z}) \, k(\boldsymbol{x}, \boldsymbol{z}).$$

Considering an approximation of the target distribution $p(s) = \sum_{i=1}^{M} \frac{1}{M} \delta_{\nu_i}(s)$, the gradient of $\mathrm{MMD}^2(q'(s), p(s))\}$ with respect to unit $\boldsymbol{\theta_i}$ is given by:

$$\nabla_{\boldsymbol{\theta_i}} \mathrm{MMD}^2(q'(s), p(s)) = -\frac{2}{N^2} \sum_{i'} \nabla_{\boldsymbol{\theta_i}} k(\boldsymbol{\theta_i}, \boldsymbol{\theta_{i'}}) + \frac{2}{NM} \sum_{j} \nabla_{\boldsymbol{\theta_i}} k(\boldsymbol{\theta_i}, \boldsymbol{\nu_j})$$

$$= -\frac{2}{N^2} \sum_{i'} \boldsymbol{\Sigma}^{-1}(\boldsymbol{\theta_i} - \boldsymbol{\theta_{i'}}) \, k(\boldsymbol{\theta_i}, \boldsymbol{\theta_{i'}}) + \frac{2}{NM} \sum_{j} \boldsymbol{\Sigma}^{-1}(\boldsymbol{\theta_i} - \boldsymbol{\nu_j}) \, k(\boldsymbol{\theta_i}, \boldsymbol{\nu_j}).$$

The update for unit $\boldsymbol{\theta_i}$ is therefore given by:

$$\boldsymbol{\theta_i} \leftarrow \boldsymbol{\theta_i} - \alpha \nabla_{\boldsymbol{\theta_i}} \mathrm{MMD}^2(q'(s), p(s)),$$

where $\alpha$ is the learning rate. As an aside, these dynamics correspond to a so-called *blob method* in the particle optimization framework (Chen et al., 2018).

The DNL Wasserstein regularization enforces the conservation os the population geometry through learning. Preserving population geometry ensures that decoders remain effective across changing contexts and stimulus distributions, thereby removing the need for retraining. To illustrate this, in addition to the simulation described in Fig.2(c), we simulate a spatial cognitive task (Krupic et al., 2015) where the environment shape changes, by removing the lower triangle as represented in Fig. 7(a,b). As observed in hippocampus place cells (Krupic et al., 2015), DNL, but not MMD, enforces the conservation of the population geometry, and therefore a linear decoder based on DNL representations adapts faster to the new distribution(Fig. 7(a-c)). In particular, we decoded the mean of the $x$ spatial coordinate of the $10\%$ closest units to the origin. Additionally we consider the covariance matrix defined by:

$$\boldsymbol{\Sigma} = \begin{pmatrix} \sigma & 0 \\ 0 & \sigma \end{pmatrix}.$$

---

[2]In this section both rewards and spatial samples are denoted by $s$.

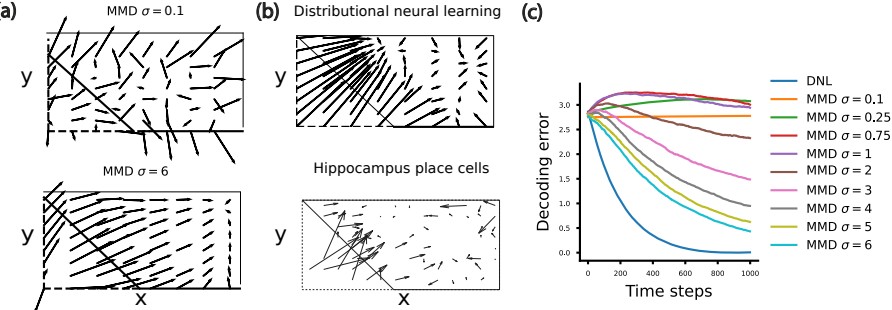

Figure 7: (a,b) Vector field generated by the MMD and DNL learning rules for the experiment where the spatial occupancy is manipulated, by removing the lower triangle (represented as dashed lines). (c) Decoding error for DNL and MMD for different bandwidths $\sigma$.

## A.3 DERIVING THE DISTRIBUTIONAL NEURAL LEARNING RULES IN DETAIL

We optimize the reward distribution representation, quantified by the KL divergence between the unit distribution and the target reward distribution, and the efficiency of the learning trajectory, quantified by the Wasserstein distance between unit distributions across iterations,

$$q_{t+1} = \arg\min_{q' \in \mathcal{Q}} \{D_{\mathrm{KL}}(q' || p(\boldsymbol{r}))) + \frac{1}{h}W_2^2(q', q_t)\}, \tag{9}$$

where $\boldsymbol{r} \in \mathbb{R}^N$, $h$ is a step-size, $p(\boldsymbol{r})$ is discretized and represented by samples from the target distribution, $q_t$ is the unit distribution at time-step $t$ and $\mathcal{Q} = \{q' : q' = \sum_{i=1}^{N} \frac{1}{N}\delta_{\boldsymbol{\theta}_i}\}$. We use the learning rules derived in Chen et al. (2018). In particular, the KL divergence can be decomposed as,

$$D_{\mathrm{KL}}(q'(\boldsymbol{r})||p(\boldsymbol{r}))) = \int_{\boldsymbol{r}} q'(\boldsymbol{r}) \log \frac{q'(\boldsymbol{r})}{p(\boldsymbol{r})} d\boldsymbol{r} = \int_{\boldsymbol{r}} q'(\boldsymbol{r})(\log q'(\boldsymbol{r}) - \log p(\boldsymbol{r})) d\boldsymbol{r} = \mathbb{E}_{q'}[\log q'] - \mathbb{E}_{q'}[\log p(\boldsymbol{r}|\boldsymbol{\theta})].$$

Introducing the likelihood function of parameters $\boldsymbol{\theta}$ given samples $\boldsymbol{s}$, $p(\boldsymbol{r}|\boldsymbol{\theta}) = l(\boldsymbol{\theta}|\boldsymbol{r})$, equation 9 (Chen et al., 2018) can be rewritten as the sum of two terms:

$$q_{t+1} = \arg\min_{q' \in \mathcal{Q}} \{ \underbrace{- \mathbb{E}_{q'}[\log l(\boldsymbol{\theta}|\boldsymbol{r})]}_{F_1} + \underbrace{\mathbb{E}'_q[\log q'] + \frac{1}{h}W_2^2(q', q_t(\boldsymbol{r}))}_{F_2}\}, \tag{10}$$

the term $F_1$ attracts the units towards the samples and the $F_2$ regularizes the unit interactions. Then, distributional neural learning (DNL) rules can be derived for each unit,

$$\boldsymbol{\theta}_i^{t+1} \leftarrow \boldsymbol{\theta}_i^t - \alpha \left( \frac{\partial F_1}{\partial \boldsymbol{\theta}_i} + \frac{\partial F_2}{\partial \boldsymbol{\theta}_i} \right) . \tag{11}$$

The gradient of $F_1$ is given by:

$$\frac{\partial F_1}{\partial \boldsymbol{\theta}_i} = -\nabla_{\boldsymbol{\theta}_i} \log l(\boldsymbol{\theta}_i|s) .$$

To approximate the gradient of $F_2$, let $\{p_{ij}\}_{i,j=1}^{N}$ denote the joint distribution (or coupling in optimal transport literature) of the unit-pair across iterations $(\boldsymbol{\theta}_i, \boldsymbol{\theta}_j^t) \in \mathbb{R}^d \times \mathbb{R}^d$. In order to obtain closed-form, explicit learning rules, an entropy penalty (weighted by $\lambda$) is added to $p_{ij}$. The term $\mathbb{E}_q[\log q]$ is minimized when the particles are uniformly distributed, i.e., when the marginal distributions $(\sum_j p_{ij})_i$ are uniform.

Combining all terms and using the definition of $W_2$, minimizing $F_2$ is equivalent to solving the following optimization problem:

$$\{p_{ij}\} = \arg\min_{p_{ij}} \sum_{i,j} \lambda p_{ij} \log p_{ij} + \frac{1}{h} p_{ij} d_{ij} \text{ such that } \sum_j p_{ij} = \frac{1}{N} \text{ and } \sum_i p_{ij} = \frac{1}{N},$$

where $d_{ij} = ||\boldsymbol{\theta}_i - \boldsymbol{\theta}_j^t||^2$. Considering the dual variables $\{\alpha_i\}_{i=1}^D$ and $\{\beta_j\}_{j=1}^D$, the Lagrangian is given by:

$$\mathcal{L}(\{p_{ij}\}, \{\alpha_i\}, \{\beta_i\}) = \left\{ \sum_{i,j} \lambda p_{ij} \log p_{ij} + \frac{1}{h} p_{ij} d_{ij} + \sum_i \alpha_i \left( \sum_j p_{ij} - \frac{1}{N} \right) + \sum_j \beta_j \left( \sum_i p_{ij} - \frac{1}{N} \right) \right\}.$$

(12)

Setting $\partial \mathcal{L}/\partial p_{ij} = 0$ yields

$$\frac{\partial \mathcal{L}}{\partial p_{ij}} = \left( \lambda \big(1 + \log p_{ij}\big) + \frac{d_{ij}}{h} \right) + \alpha_i + \beta_j = 0 \quad \implies \quad p_{ij}^* = \exp\left(-1 - \frac{d_{ij}/h + \alpha_i + \beta_j}{\lambda}\right).$$

Setting $u_i = e^{-\frac{1}{2} - \frac{\alpha_i}{\lambda}}$ and $v_j = e^{-\frac{1}{2} - \frac{\beta_j}{\lambda}}$, the gradient of $F_2$ with respect to $\boldsymbol{\theta}_i$ is given by:

$$\frac{\partial F_2}{\partial \boldsymbol{\theta}_i} \approx \sum_j 2 u_i v_j \left( \frac{d_{ij}}{\lambda} - 1 \right) e^{-d_{ij}/\lambda} (\boldsymbol{\theta}_i - \boldsymbol{\theta}_j^t),$$

where we considered the $\lambda$ parameter absorbs the step-size $h$. Theoretically, the dual values $\{\alpha_i, \beta_j\}$ can be computed using Sinkhorn's fixed point iteration (Algorithm 1 in (Cuturi, 2013)). In practice, we use a constants scalar $c$ to approximate $2 u_i v_j$.

### A.4 DERIVING THE 1-DIMENSIONAL APPROXIMATION TO EXPECTILE LEARNING RULES IN DETAIL

In the limit $\lambda \to \infty$, the $F_2$ gradient with respect to $\theta_i$ tends to:

$$\frac{\partial F_2}{\partial \theta_i} \to -\sum_j c(\theta_i - \theta_j^t) = \sum_j c(\theta_j^t - \theta_i).$$

Since in one dimensions, the units can be ordered from lower to higher values:

$$\frac{\partial F_2}{\partial \theta_i} = \sum_{j : \theta_j < \theta_i} c(\theta_j^t - \theta_i) + \sum_{j : \theta_j > \theta_i} c(\theta_j^t - \theta_i).$$

By approximating the interactions with a given unit $\theta_i$ in $F_2$ by the interactions with the mean of the units to its left ($\bar{\theta}_{\text{LEFT}}$) and right ($\bar{\theta}_{\text{RIGHT}}$), we get:

$$\frac{\partial F_2}{\partial \theta_i} \approx c N_{\text{LEFT}}(\bar{\theta}_{\text{LEFT}} - \theta_i) + c N_{\text{RIGHT}}(\bar{\theta}_{\text{RIGHT}} - \theta_i),$$

where $N_{\text{LEFT}}$ and $N_{\text{RIGHT}}$ are the number of units to the left and right of $\theta_i$ respectively. Since $\tau = \frac{N_{\text{LEFT}}}{N}$ fraction of the units are to the left and $1 - \tau$ are to the right:

$$\frac{\partial F_2}{\partial \theta_i} \approx +c\tau(\bar{\theta}_{\text{LEFT}} - \theta_i) + c(1 - \tau)(\bar{\theta}_{\text{RIGHT}} - \theta_i).$$

Noticing $\bar{\theta}_{\text{LEFT}} \approx \mathbb{E}[r | r \le \theta_i]$ and $\bar{\theta}_{\text{RIGHT}} \approx \mathbb{E}[r | r > \theta_i]$, considering the stochastic approximation $\bar{\theta}_{\text{LEFT}} \approx r$ when $r \le \theta_i$ and $\bar{\theta}_{\text{RIGHT}} \approx r$ when $r > \theta_i$:

$$\frac{\partial F_2}{\partial \theta_i} \approx \tau c \delta_{r < \theta_i}(r - \theta_i) + (1 - \tau) c \delta_{r > \theta_i}(r - \theta_i).$$

Considering a Gaussian likelihood with standard deviation $\beta$, $l(\theta_i | r) \sim \mathcal{N}(r, \beta^2)$, the $F_1$ term is given by:

$$\frac{\partial F_1}{\partial \theta_i} = -\frac{1}{\beta^2}(r - \theta_i).$$

Setting $c = \beta = 1$ our learning rules become:

$$\theta_\tau \leftarrow \theta_\tau + (1 - \tau)\delta_{r < \theta_i}(r - \theta_i) + \tau \delta_{r > \theta_i}(r - \theta_i),$$

which are the one-dimensional expectile learning rules (Bellemare et al., 2017).

## A.5 DERIVING SUFFICIENT CONDITIONS FOR OPTIMAL DISTRIBUTIONAL LEARNING TRAJECTORY

In this section we provide sufficient conditions for the DNL learning rules to converge to globally optimal transport map from the initial to the converged distribution and show that this condition is always satisfied for one-dimensional distributions.

The Jordan–Kinderlehrer–Otto (JKO) flow (Jordan et al., 1998) states that the trajectory obtained by solving the Fokker-Planck equation (FPE) is the gradient flow of the appropriate target objective (e.g. KL-divergence between initial and target densities) in the Wasserstein space of probability measures Jordan et al. (1998); Otto (2001). We can therefore draw on tools from FPE convergence analysis (Kim & Milman, 2012), to characterize the convergence of our distributional neural learning rules.

The FPE does not minimize in general the global Wasserstein metric Lavenant & Santambrogio (2022); Tanana (2021). Here, we revisit a classic result that provides sufficient conditions for the minimization of the global Wasserstein metric Kim & Milman (2012).

The FPE describes the time-evolution of the probability density $\rho$ of the random vector $R$

$$\frac{\partial \rho}{\partial t} = \nabla \cdot (r\rho) + \Delta\rho. \tag{13}$$

It generates a curve $(\rho_t)_{t \geq 0}$ of probability measures that converge to the unit standard Gaussian $\gamma$. Building on theoretical advances that connect transportation of measures to the FPE Jordan et al. (1998), Eqn. 13 can be recast as the transport equation

$$\frac{\partial \rho(r)}{\partial t} = -\nabla \cdot (\rho v), \tag{14}$$

where the velocity field is given by

$$v(t, r) = -r - \nabla \log \rho_t(r), \tag{15}$$

and can be interpreted as the Wasserstein velocity of the curve $(\rho_t)_{t \geq 0}$ (i.e. the vector field providing the steepest descent in functional space according to the Wasserstein metric) Santambrogio (2015). We consider the ordinary differential equation (ODE) generated by $v$ and define a function of transport maps $S : \mathbb{R}_{\geq 0} \times \mathbb{R}^{d \times d} \to \mathbb{R}^{d \times d}$ defined by

$$\begin{cases} \frac{\partial S}{\partial t} = v(t, S) = -S - \nabla \log \rho_t(S), \\ S(0, .) = Id. \end{cases} \tag{16}$$

This allows us to view $\rho_t$ as the pushforward of $\rho_0$ under the flow map $S_t$, $\rho_t = S_t \# \rho_0$. We will derive sufficient conditions for $S_\infty$ to be the optimal transport map between $\rho_0$ and $\gamma$ and minimize the global 2-Wasserstein metric. Differentiating Eqn. 16 we obtain

$$\frac{\partial DS_t}{\partial t} = -DS_t - D^2 \log \rho_t(S_t)DS_t = -(I + D^2 \log \rho_t(S_t))DS_t \quad . \tag{17}$$

To simplify notation let $B_t(r) = -(I + D^2 \log \rho_t(S_t))$. If all $D^2 \log \rho_t(S_t)$ commute, then all $B_t(r)$ also commute and $DS_t$ remains symmetric along the flow and we can write

$$DS_t(x) = \exp\left(\int_0^t B_\tau(x)d\tau\right), \tag{18}$$

from which it follows that $DS_t$ is pointwise semi-definite and hence $S_t$ must be the gradient of a convex function. Finally, taking in account Brenier's theorem Santambrogio (2015) and taking the limit $t \to \infty$, we conclude $S_\infty$ is the optimal transport map.

Conversely, this result states that probability measures that generate flows with symmetric Jacobians imply that the learning trajectory will be equivalent to the optimal transport map, i.e. Eqn. 13 minimizes the 2-Wasserstein metric. This criterion is satisfied by radially symmetric probability measures such as Gaussians with isotropic covariance. In SFig. 8c we give an example, where the initial and target distributions are radially symmetric, hence the flow Jacobians are symmetric and Eqn. 13 coincides with the optimal transport. Notably, the Jacobians along the gradient flow of a one-dimensional probability measure is scalar and thus is trivially symmetric. Therefore, distributional RL

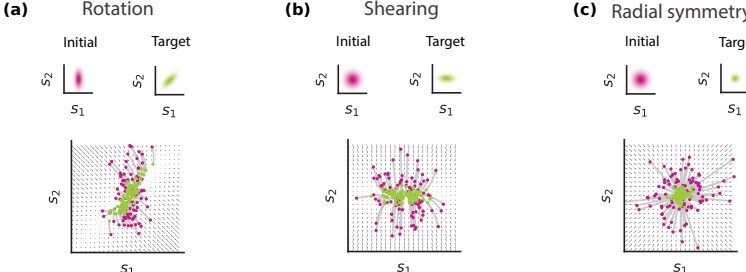

Figure 8: In (a) and (b) we simulate cases where the Jacobian of the unit update rules is not symmetric and therefore generates globally *twisted* transport maps. In (c) we simulate the case where the initial and target distributions are radial symmetric, and therefore the Jacobian of the update is symmetric and the transport map is optimal. Each panel represents on the top the initial (magenta) and target (green) distributions and on the bottom the learning trajectory, with the gradient flow represented as black arrows.

(without function approximation) converges on the optimal learning trajectories in the large sample limit.

However, in general this implies that, Eqn. 13 flow introduces a *curl* on the trajectory of probability measures. For example, if we replace $-r$ by $-Ar$ in the drift in Eqn. 15, then $\rho_t$ converges to a Gaussian with covariance $A^{-1}$. Selecting a matrix $A$ that does not commute with the initial density covariance, then Eqn. 13 generates curled trajectories, such as the rotation and shearing examples shown in SFig.8a,b).

## A.6    DEEP DISTRIBUTIONAL NEURAL LEARNING EXPERIMENTS

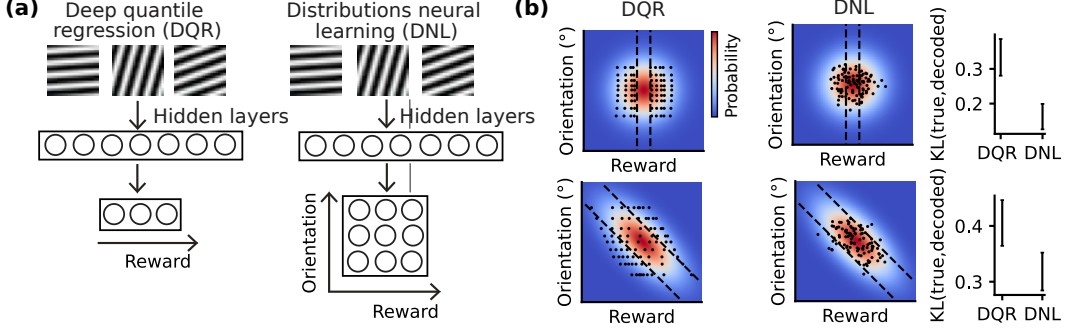

Figure 9: (a) Representation of the DQR and DNL networks. (b) The dots correspond to a quantile for a given orientation in the DQR case and a multidimensional quantile in reward and orientation in the DNL case. The target distributions are represented in the background. On the right, the 95% C.I. of KL divergence between the quantiles and the target distribution is shown for 10 runs.

We train a deep quantile regression network (DQR) Dabney et al. (2018b) to predict $N = 100$ reward quantiles and a deep distributional neural learning network (DNL) to predict $N = 100$ multidimensional quantiles over reward magnitude and orientation. The DQR and DNL networks have two hidden layers with 1024 and 256 units. The Adam optimizer was used, with an initial learning rate of 0.0001. Mini-batches of 200 samples were used, and the networks were trained for 1000 epochs. Since the Wasserstein distance is computationally expensive, we employ the Maximum Mean Discrepancy (MMD) as a practical approximation in these simulations Gretton et al. (2012). For the MMD computation, we set the kernel bandwidth parameter bw = 4 and used 1000 samples to estimate the distributions.

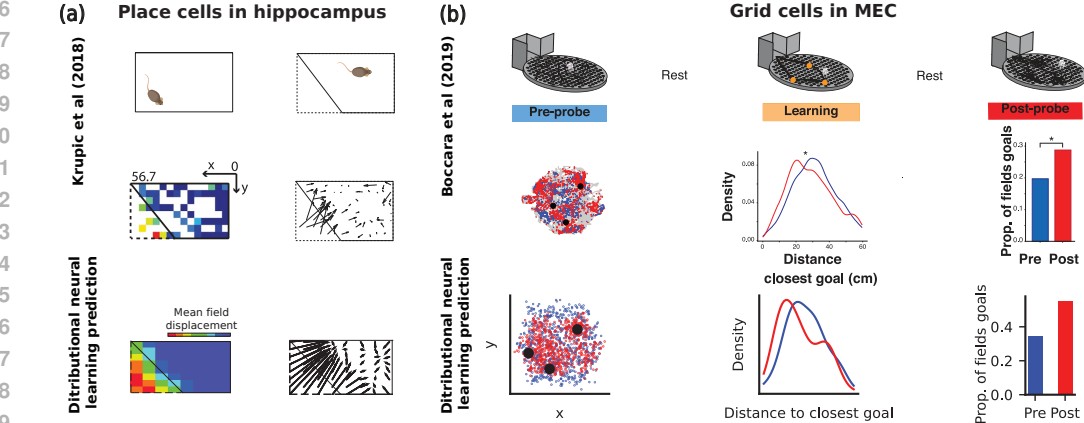

Figure 10: (a) Top: adapted from Krupic et al. (2018). Middle: Mean field displacement (left) and vector field (right) for an example place cell population, when modifying the shape of the environment. Bottom: predictions from DNL theory. (b) Top: adapted from Boccara et al. (2019). Middle : Spikes of an example grid field (left), distribution of the distance to the closest goal (center) and proportion of fields at goal (right) pre (blue) and post-probe (red) to new reward locations. Bottom: predictions from DNL theory.

### A.7 MODELING CELL ADAPTATION IN THE BRAIN VIA DISTRIBUTIONAL NEURAL LEARNING

In order to demonstrate the utility of the distributional neural learning framework for understanding neural coding in the brain, we model two datasets of neural population codes in spatial navigation paradigms whereby the target stimulus is a two-dimensional representation of space.

Krupic et al Krupic et al. (2018) allowed rodents to explore an environment before exposing them to a locally transformed version of it (SFig. 10 (a), top) and found that place cells in the hippocampus adapted their tuning to the altered shape of the environment (SFig. 10 (a), middle). We modeled each place cell tuning in terms of each neuron's preferred spatial position $\{\theta_i\}_{i=1}^N$. The spatial distribution $p_\theta$, optimized for this population, was interpolated from $\{\theta_i\}_{i=1}^N$, conceptualized as approximate Dirac delta functions in probability space. Initially, samples of $x$ and $y$ spatial positions that uniformly covered the rectangular space were given until convergence. Subsequently, only samples from the trapezoidal region were provided. Our simulations show that the DNL theory generates place fields that adapt to the shape of the environment (SFig. 10 (a), bottom).

In another study, Boccara et al Boccara et al. (2019) trained rats to daily learn three new reward locations on a cheeseboard maze while recording from the medial entorhinal cortex (MEC) and found many grid fields moved toward goal location (SFig. 10 (b), middle). We modeled the tuning functions of grid cells as two-dimensional hexagonal grids. Samples of reward $x$ and $y$ spatial positions were given. Our simulations show that the DNL theory generates grid fields that move towards reward locations (SFig. 10 (b), bottom).

# B EXPERIMENTAL DETAILS

## B.1 DISTRIBUTIONAL NEURAL LEARNING CONVERGENCE SIMULATIONS

For the DNL simulation shown in Fig. 2b (main text), we initialized the $N = 100$ units uniformly spaced in the rectangle delimited by $x = 3.5$ and $x = 5.75$ and $y = 0$ to $y = 10$. The following hyperparameters were used for the distributional neural learning rules defined in Eqn. 6 of the main manuscript:

- Total number of iterations: 1000.
- Batch size: 500.
- Gaussian KDE bandwidth: 0.4.
- Unit interaction regularization parameter: $c = 1000$
- Interaction attraction-repulsion term $\lambda = 0.1$.
- Learning rate $\alpha = 0.1$ reduced by $50\%$ every 500 iterations.

For the DNL simulation shown in Fig. 2c (main text) we initialized $N = 12$ units sampled from a Gaussian with mean $\mu = [1, 1]$ and covariance matrix $\Sigma = \begin{bmatrix} 0.25 & 0 \\ 0 & 0.25 \end{bmatrix}$. Samples from a Gaussian were given with mean $\mu = [3, 3]$ and covariance matrix: $\Sigma = \begin{bmatrix} 0.5 & 0 \\ 0 & 0.5 \end{bmatrix}$. The likelihood function was estimated from the stimulus samples using kernel density estimation, as implemented in the gaussian kernel density function from SciPy Virtanen et al. (2020). The following parameters were used for the distributional neural learning rules defined in Eqn. 6 of the main manuscript:

- Total number of iterations: 30 000.
- Batch size: 10.
- Gaussian KDE bandwidth: 3.
- Unit interaction regularization parameter $c = 200$.
- Interaction attraction-repulsion term: $\lambda = 0.1$.
- Learning rate $\alpha = 0.1$.

For the MMD, the total number of iterations and batch size were kept the same. To ensure that the absolute update magnitude was not smaller than in DNL, we set the learning rate to $\alpha = 2$. Gaussian kernels with different covariance matrix were used:

$$\Sigma = \begin{bmatrix} \sigma & 0 \\ 0 & \sigma \end{bmatrix},$$

with $\sigma = 0.1, 0.25, 0.5, 0.75$. The value decoded error was the square of the difference of the true and decoded value, $V = \sum_i r_i \gamma^{t_i}$, where $\boldsymbol{t_i} = (t_i, r_i)$, computed using the $25\%$ units closest to the origin.

For the simulations of the spatial cognitive task the following parameters were considered:

- Total number of iterations: 1000.
- Batch size: 25.
- Gaussian KDE bandwidth: 0.4.
- Unit interaction regularization parameter $c = 5$.
- Interaction attraction-repulsion term: $\lambda = 0.07$.
- Learning rate $\alpha = 0.001$.

In this case, the mean of the $x$ spatial coordinate of the $10\%$ closest units to the origin was decoded. To ensure that the absolute update magnitude was not smaller than in DNL, we set the learning rate to $\alpha = 4.5$.

### B.2 Modeling cell adaptation via distributional neural learning

For the adaptation of place cells (Fig. 10a), we considered the following parameters for the distributional neural learning rules defined in Eqn. 6 of the main manuscript:

- Total number of iterations: 500 .
- Batch size: 25 .
- Gaussian KDE bandwidth: 0.4 .
- Unit interaction regularization parameter: $c = 5$ .
- Interaction attraction-repulsion term: $\lambda = 0.07$ .
- Learning rate $\alpha = 0.001$ .

For the adaptation of grid cells (10b) we considered the following parameters for the distributional neural learning rules defined in Eqn. 6 of the main manuscript:

- Total number of iterations: 350 .
- Batch size: 25 .
- Gaussian KDE bandwidth: 0.4 .
- Unit interaction regularization parameter: $c = 1000$ .
- Interaction attraction-repulsion term $\lambda = 0.08$ .
- Learning rate $\alpha = 0.01$ .

### B.3 Generalization in RL over reward dynamics using time-magnitude maps

We use two different environments. The first uses $n$ states (or patches) and $n$ distinct stimuli, each associated with a different reward delay and magnitude following the stimulus. The second is a gridworld environment with similarly structured stimuli and rewards. A stimulus appears with a random probability at each time step with equal probability for all stimulus.

These experiments were designed to demonstrate that a reinforcement learning agent can learn structure in the environment rapidly using the reward TMD. To fairly compare learning to a standard RL model, in the training steps, random actions were taken for both the standard RL and TMRL models. This was done just to be certain that the convergence rates were not due to bad policy initializations or local minima. During test steps, both agents used a greedy policy.

#### B.3.1 Patch experiments

This simple environment was designed such that the Q-value of actions (one possible action for every possible state transition) only depended on the probability of reward at the next time step. Tracking the reward delays and magnitudes is shown to be useful but there is no need to learn a policy for sequential decisions as an action from one state can bring the agent to any other state in the next time step. As long as the agent is able to track the time since each cue and has learned a mapping from the time since each cue to a probability of reward appearing with a certain magnitude, the agent can simply take the action to the state with the highest expected reward at the next time.

The reward rates shown are during test trials of 10000 time steps using a greedy policy in both models. The parameters are given below:

- number of runs for each model: 10
- number of states/stimuli: $n \in \{2, 3, 4, 5\}$
- maximum reward delay: 5
- maximum reward magnitude: 4
- probability of each stimulus shown at every time step: 0.1
- number of training steps: 10000
- number of test steps: 10000
- test every 100 steps of training

**Patch environment**   The environment has $n$ patches/states, $n \in \{2, 3, 4, 5\}$. The agent can perform an action to transition from its current state to any other deterministically. The environment also has the same number $n$ of stimuli which each cue a reward after a certain delay and magnitude drawn from a distribution. The reward cued by stimuli $i$ would be available in state $i$ for simplicity but similar results would hold without this constraint. At every time step, regardless of the current state of the agent, the probability of stimuli $i$ appearing was drawn with probability 0.1, independent of all other stimuli. Multiple stimuli were allowed to appear at the same timestep. A second stimulus was also allowed to occur before the delivery of reward from a previous stimulus. Each stimulus was associated with a probabilistic reward TMD which was constructed randomly. For each stimulus, the TMD was constructed to have 0.5 probability of reward appearing at some delay $t_1$ after the stimulus with magnitude $r_1$ and 0.5 probability for delay $t_2$ and $r_2$. For each stimulus $t_1, t_2, r_1$ and $r_2$ were randomly chosen with uniform probability with maximum magnitude 4 and maximum delay 5. After every presentation of a stimulus $i$, a reward would be available at state $i$ with delay and magnitude drawn from the true TMD $i$, independently after each presentation. The reward would then only be available at only the time after the cue and would not remain after that time step.

**TMRL agent**   This agent separately tracked two parts of its state, the current patch it was in $s_t$ and the time since each of the stimuli $t_i$ (up to the maximum delay time). At every time step, if a stimulus $i$ was seen, the current time delay $t_i^t$ was reset to 0 for that stimulus, otherwise the time increased by 1.

The reward TMD ($\text{TMD}_i(s', t', r')$ is the probability of reward at location $s'$ of magnitude $r'$ at delay time $t'$ after stimulus $i$) was learned by the agent using the DNL learning rules, $\text{TMD}_i(s, t, m) \approx \frac{1}{N} \sum_i \delta_{\boldsymbol{\theta}_j^s}(r) \delta_{\boldsymbol{\theta}_j^s}(t)$ using units $\boldsymbol{\theta}_j = (t_{\boldsymbol{\theta}_j}, r_{\boldsymbol{\theta}_j})$. To make the Q value easy to compute, the approximation discretized the magnitude and delay space into a matrix and the percentage of units in each discrete square was used as a probability. Parameters for the DNL learning are given below:

Parameters of the TMRL agents including the ablations:

- Batch size: 100.
- Gaussian KDE bandwidth: 0.5.
- Unit interaction regularization parameter: $\gamma = 100.0$.
- Interaction attraction-repulsion term $\lambda = 4.0$.
- DNL learning rate starting at $\alpha_{DNL} = 0.001$ and decreased to $\alpha_{DNL} = 0.0001$ over 5000 training steps
- Number of units $N = 25$.
- Covariance of 0.05 times the identity matrix.
- TMD learning rate $\alpha_{TMD} = 0.01$

At test time, the agent chooses greedy actions by looking only at the reward TMDs for each stimulus and the current time since each stimulus $t_i$. Using this, it can construct the probability of reward times and magnitudes for each state ($TMD_{total}^t$). Here we simply update $TMD_{total}^t$ at every time step given all the observed stimuli $j$ at time $t$,

$$TMD_{total}^t(s, t', r') \quad \leftarrow \quad TMD_{total}^{t-1}(s, t'+1, r') + \sum_{j \text{ observed at time } t} TMD_j(s, t', r')$$

$$\text{for } 1 \leq t' \leq t_D - 1 \tag{19}$$

$$TMD_{total}^t(s, t', r') \quad \leftarrow \quad \sum_{j \text{ observed at time } t} TMD_j(s, t', r') \qquad \text{for } t' = t_D \tag{20}$$

where $t_D$ is the maximum reward delay. The greedy policy simply picks the action for the state with the largest expected reward at the next time step

$$a_t = \text{argmax}_i \sum_r r \, TMD_{total}^t(s = i, 1, r) \tag{21}$$

The full algorithm is given in Algorithm 1.

---

**Algorithm 1** TMRL Agent in Patch Environment

---

1: **Initialize** $TMD_i$, $\forall i$ to all 0
2: **Initialize** $t_i$ to $t_D$, $\forall i$

3: **function** ACT($s$)
4:     Choose action $a_t$ from $TMD_{total}^t$ using greedy policy          ▷ Equation 21
5: **end function**

6: **function** TRAIN($s, a, r, s'$)
7:     **if** stimulus j is observed **then**
8:         $t_j^t = 0$
9:     **else**
10:        $t_j^t \mathrel{+}= 1$
11:     **end if**
12:     **if** reward is observed **then**
13:         **for** i from 0 to $C$ **do**          ▷ C is the number of stimuli
14:            **if** $t_i^t < t$ **then**
15:               $TMD_i(\cdot, \cdot, \cdot) \leftarrow TMD_i(\cdot, \cdot, \cdot) + \alpha \delta_{TMD_i}(\cdot, \cdot, \cdot)$     ▷ Equation 5
16:            **end if**
17:         **end for**
18:     **end if**
19: **end function**

---

**Standard RL agent**   The standard RL agent had an expanded state space that represented the current patch and the time since each stimuli. This state space has a unique, discrete state for every possible combination of cue delays (up to the maximum delay time). The state space is constructed this way to ensure that the Markov property is satisfied.

Value for this agent was learned by standard temporal difference learning

$$\delta_V(t) \quad = \quad r_t - V(s_t) \tag{22}$$
$$V(s_t) \quad = \quad V(s_t) + \alpha \delta_V(t) \tag{23}$$

where $r_t$ is the reward at time $t$. Here we do not need to include $\gamma V(s_{t+1})$ because there is no need for sequential decisions. We just learn a value for every state that is the average reward.

As the actions are deterministic and because we only give reward when the agent is in the correct state, we can determine the action $a_t$ under the greedy policy at state $s_t$ by just looking at the value of $V(s_{t+1})$ after taking that action. In the patch environment this is particularly easy as each action $a_t = i$ corresponds to moving to one of the locations $i$. However, the state space is a combination of state and delay so $s_{t+1}$ should be the next location $i$ and the delays from stimuli after one time step.

The full algorithm is given in Algorithm 2

---

**Algorithm 2** Standard Agent in Patch Environment

---

1: **Initialize** $V(s)$
2: **Initialize** state $s$

3: **function** ACT($s$)
4:     Compute $Q(a) = V(s')$ for actions $a$ from state $s$ and transitioning to state $s'$
5:     **return** $\arg\max_a Q(a)$
6: **end function**

7: **function** TRAIN($s, a, r, s'$)
8:     $V(s) \leftarrow V(s) + \alpha \left[ r_t - \gamma V(s) \right]$          ▷ Treat as bandit task at $s$
9: **end function**

---

**Ablation experiments**   In order to demonstrate the necessity of the multidistributional distribution in TMRL in achieving fast generalizations to different stimulus combinations in the task, we perform an ablation experiment. We learn on the patch environment with the same TMRL parameters.

However, during testing, when using the learned stimulus TMD to compute the action, we only use the time dimension and the expected value along the magnitude dimension (time only) or we only use the magnitude dimension and expectation along the time dimension (magnitude only) of the distributions. Here, the agents have the same structure but only compute policy with 1D distributions rather than the joint. To highlight the differences, we use stimulus TMDs in the environment where the TMD for stimulus $i$ has 0.5 probability of reward at $(t_i^1, r_i^1)$ and 0.5 probability of reward at $(t_i^2, r_i^2)$. For every stimulus $i$, $t_i^1$ and $t_i^2$ had a difference of 3 time steps and $r_i^1$ and $r_i^2$ had a magnitude difference of 8. In this way, the expectation across either time or magnitude of the learned TMDs were poor representations of the distribution. Furthermore, we simulated these environments with different values of the probability of a stimulus appearing. As seen in Supplementary Fig. 11, the magnitude only agent performs extremely poorly. This is because timing information in crucial in this environment as rewards are only available at delay $t_i^1$ or $t_i^2$ after stimulus i. The magnitude only agent expects the reward at the expectation over delays when there is no reward. The time only agent performs fairly well with only the delay distribution. This agent can go the the correct states at the correct times. However, as the probability of stimuli increase, it becomes more likely that rewards occur at the same time in different states. In these situations, the time only agent must act using only the expected value of the magnitudes rather than comparing $r_i^1$ with $r_j^1$ or $r_i^2$ with $r_{j1}$ etc. The expected value fails to represent all the information available to the full TMRL agent. Parameters of the environment are listed below:

- number of runs for each model: 10
- number of states/stimuli: 3
- maximum reward delay: 5
- maximum reward magnitude: 10
- probability of each stimulus shown at every time step: 0.1
- number of training steps: 100000
- number of test steps: 10000
- test every 100 steps of training
- cue probability: $\{0.05, 0.1, 0.2\}$

Parameters of the TMRL agents including the ablations:

- Batch size: 100.
- Gaussian KDE bandwidth: 0.5.
- Unit interaction regularization parameter: $\gamma = 100.0$.
- Interaction attraction-repulsion term $\lambda = 4.0$.
- DNL learning rate starting at $\alpha_{DNL} = 0.001$ and decreased to $\alpha_{DNL} = 0.0001$ over 5000 training steps
- Number of units $N = 25$.
- Covariance of 0.05 times the identity matrix.
- TMD learning rate $\alpha_{TMD} = 0.01$

### B.3.2 GRIDWORLD EXPERIMENTS

This environment was designed such that sequential decisions are necessary as agents must travel to the next reward location and must know if they will reach that location before the delay time. If two stimuli appear close in time, the agent must decide which of the different reward locations will give the best expected reward, again considering the reward magnitudes and delays with travel time to the location.

During training, we used random actions to collect observed samples for the TMRL standard RL, and QR-RL agents. This was done simply to show that learning speed was not due to better initial policies of the model. The reward rates shown are during test trials of 10000 time steps using a greedy policy in all models. The parameters are given below:

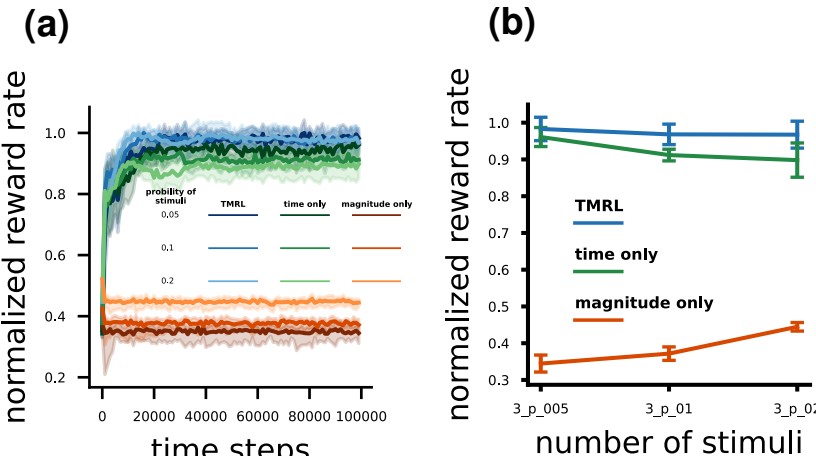

Figure 11: **Ablation experiments** TMRL (blue) using DNL update rules compared to time only (green) and magnitude only (red) ablated TMRL agents. Performance in the patches environment for a varying probability of appearance of the stimuli. (a) Reward rate of agents training over time steps on the patch environment. Reward rate is normalized to the largest mean reward achieved so that performance can be compared across environments with different probability of stimuli appearance. (b) Same data as (a) but only showing the final reward rate after 100000 train steps.

- number of runs for each model: 10
- gridworld size: $5 \times 5$
- gridworld number of possible actions: 4
- maximum reward delay: 5
- maximum reward magnitude: 4
- probability of each stimulus shown at every time step: 0.1
- number of cues: $n \in \{3, 4, 5\}$
- number of training steps: 300000
- number of test steps: 10000
- test every 1000 steps of training

**Gridworld environment**    The gridworld was constructed with a grid of size $5 \times 5$ with 4 possible deterministic actions: up, down, left, and right. Similar to the patch environments, this gridworld environment also has $n$ stimuli, $n \in \{2, 3, 4, 5\}$, each associated with a TMD constructed the same way with maximum magnitude 4 and maximum delay 5. In addition, the location of the reward on the grid was also randomly chosen for each stimulus. Again, the probability of observing a stimulus at each timestep was a fixed probability of 0.1 for every stimulus. Just as in the patch environment, we place no restriction on when the stimuli can appear. Multiple can appear simultaneously, they can appear before the reward was given for a previous stimulus, and they can appear while the agent is in any state. The reward is only available at a certain time and location, with a certain magnitude, and is not available afterward if not collected.

**TMRL agent**    Similar to the previous TMRL agent, this agent separately tracked two parts of its state, the current location in the gridworld $s_t$ and the time since each of the stimuli $t_i$ (up to the maximum delay time $t_D$). These were updated using the same method as above.

The reward TMD was also learned similarly by the agent through temporal difference learning with $s$ now the location on the grid. The $TMD_i$ for stimulus $i$ was learning using the DNL update rules in equation 5 and discretized to create a matrix in order to use the successor representations.

We again find the $TMD_{total}^t$ that tracks the possible reward locations, delay from current time t, and magnitude given the past history of observed stimuli as in equation 19.

To select actions using the learned TMD, the TMRL agent also learns $n$ successor representations (SR). In these experiments, we just set $n$ to the number of stimuli. Each SR $M^{\pi_i}$ will represent the discounted future state occupancy of state $s'$ in the grid starting from state $s$ and following a policy $\pi_i$. Multiple SR's allow the agent to learn multiple policies and combine them using generalized policy improvement (GPI) Barreto et al. (2016). This is useful when the agent encounters a combination of stimuli with different timings and must find the policy that maximizes reward.

In the SR framework, the $TMD^t_{total}$ will serve as our reward vector in the vanilla SR. Essentially, given the past history of observed stimuli, the $TMD^t_{total}$ tracks the location of possible rewards, their time delay from the current time point, and their magnitude. To use the TMD with the SR, we simply expand the SR matrix to include the time of occupancy of each state, $M^\pi(s, s', t)$, where $t$ is the number of time steps after starting in state $s$ (see Supplementary Fig. 14). We only track $t$ up to the maximum delay time $t_D$ since this is the time horizon we care about. Then multiplying the SR $M^\pi(s, s', t)$ by the $TMD^t_{total}(s', t, r)$ gives us the discounted probability of seeing reward of magnitude $r$ starting from state $s$ and following policy $\pi$.

First we find a weighting for the update of SR given by $W_i(s_t, a_t)$ for each policy $\pi_i$

$$
\begin{aligned}
H_i(s', t - t_i, m) &= TMD_i(s', t, r) \qquad \text{for } t_D \geq t_i \qquad\qquad (24)\\
W_i(s_t, a_t) &= \sum_r r\, H_i(s_{t+1}, 0, r) + \sum_{s', t', r'} r'\, M^{\pi_i}(s_{t+1}, s', t') \cdot H_i(s', t'+1, r') \qquad (25)
\end{aligned}
$$

where $a_t$ is action, $s_t$ is state/location at time t, $s_{t+1}$ is the next state after taking action $a_t$, and $t_i$ is the time since the previous observation of stimulus $i$. $H_i$ is the reward time and magnitude distribution for stimulus $i$ given the current time from the previous observation of stimulus $i$, $t_i$. $t_D$ is the maximum reward delay that is tracked. $H_i$ is similar to $total\_TMD$ but only for each stimulus separated. We use this only to get importance weights $W_i(s_t, a_t)$ used in the updates of the SR as seen below. These importance weights bias the SR to represent separate policies for each stimulus which can later be combined for GPI.

To learn this SR, we again use TD learning but weight the updates such that the policy $\pi_j$ with the maximum action probability for the action $a_t$ that was taken gets larger weight.

$$
\begin{aligned}
j &= \max_i W_i(s_t, a_t) & (25)\\
w(j) &= 1 - \epsilon & (26)\\
w(i) &= \epsilon/A, \forall i \neq j & (27)\\
\delta_{M_l}(s', t) &= \mathbb{1}[s' = s_{t+1}, t = 0] + \gamma * M^{\pi_l}(s_{t+1}, s', t+1) - M^{\pi_l}(s_t, s', t) & (28)\\
M^{\pi_l}(s_t, s', t) &\leftarrow M^{\pi_l}(s_t, s', t) + \alpha w(l)\delta_{M_l}(s, s', t) & (29)
\end{aligned}
$$

where $\epsilon = 0.1$ is a free parameter and $A = 4$ is the number of possible actions.

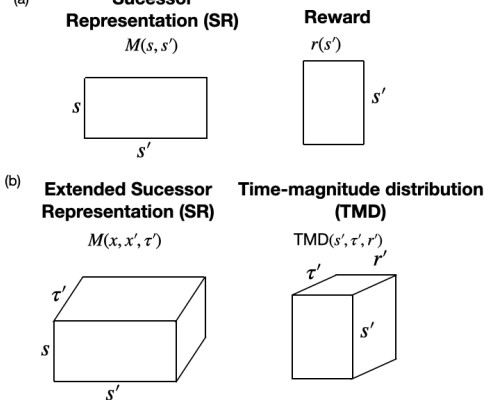

Figure 12: (a) In order to compute value in a state $s$, the SR over future states is multiplied by the reward vector. (b) We generalize this approach by considering the SR is also defined over future times, so that it may be combined with the TMD representations.

Finally, to find the best action, we use GPI.

$$
\begin{aligned}
Q_j(s_t, a_t) &= \sum_r r \, TMD^t_{total}(s_{t+1}, 0, r) + \\
&\qquad \sum_{s',t,r'} r' \, M^{\pi_j}(s_{t+1}, s', t) \cdot TMD^t_{total}(s', t+1, r') \qquad (30) \\
a_t &= \operatorname{argmax}_{a_t} \max_j Q_j(s_t, a_t) \qquad (31)
\end{aligned}
$$

The full algorithm is given in Algorithm 3.

---

**Algorithm 3** TMRL Agent in GridWorld

---

1: **Initialize** $TMD_i$, $\forall i$ to all 0
2: **Initialize** $t_i$ to $t_D$ $\forall i$

3: **function** ACT($s$)
4:      Choose action $a_t$ from $TMD^t_{total}$ and $M^{\pi_i}$ using GPI            ▷ Equation 37 - 38
5: **end function**

6: **function** TRAIN($s, a, r, s'$)
7:      **if** stimulus j is observed **then**
8:          $t_j = 0$
9:      **else**
10:        $t_j \mathrel{+}= 1$
11:     **end if**
12:     Update $TMD^t_{total}$                                       ▷ Equation 19
13:     **if** reward is observed **then**
14:        Update $TMD_i$ using DNL learning rules
15:     **end if**
16:     **for** i from 0 to $C$ **do**                           ▷ C is the number of stimuli
17:        $M^{\pi_i}(s_t, s', k) \leftarrow M^{\pi_i}(s_t, s', k) + \alpha w(i)\delta_{M_i}(s, s', k)$     ▷ Update SR. Equation 25
18:     **end for**
19: **end function**

---

**Standard RL agent**    This agent simply used standard Q-learning as shown in Algorithm 4. The state $s$ is a state that represents both the current location on the grid but also the current time delay since each of the stimuli. The state space is constructed this way to ensure that the Markov property is satisfied.

---

**Algorithm 4** Standard Agent in GridWorld

---

1: **Initialize** $Q(s, a)$ for all states $s$ and actions $a$

2: **function** ACT($s$)
3:      Choose action $a$ from $s$ using greedy policy on $Q$
4: **end function**

5: **function** TRAIN($s, a, r, s'$)
6:      $Q(s, a) \leftarrow Q(s, a) + \alpha \left[ r + \gamma \max_{a'} Q(s', a') - Q(s, a) \right]$
7: **end function**

---

**Quantile Regression Reinforcement Learning (QR-RL) agent**    The QR-RL agent uses the same expanded state space over grid location and time since each stimulus as the standard RL agent but also learns the return distribution using quantiles rather than the expected value. This is to show that simply adding the magnitude distribution is not enough to see any generalization performance increase. For a comparison with only the grid location as a state space, we use the ablation experiments in Supplementary Section B.3.1. The algorithm for this model is given in 5. The parameters are the same as above but with $N = 50$ quantiles used and huber $\kappa = 1$.

---

**Algorithm 5** Distributional RL with Quantile Regression

---

1: **Initialize** quantiles $\theta(s, a) \in \mathbb{R}^N$ for all states $s$, actions $a$
2: **Set** quantile midpoints $h_i = \frac{i+0.5}{N}$, $i = 0, \ldots, N - 1$

3: **function** ACT($s$)
4:     Compute mean action-values $Q(a) = \frac{1}{N} \sum_{i=1}^{N} \theta_i(s, a)$
5:     **return** $\arg\max_a Q(a)$
6: **end function**

7: **function** TRAIN($s, a, r, s'$)
8:     $\theta \leftarrow \theta(s, a)$                                    ▷ Current quantiles
9:     $a' \leftarrow$ ACT($s'$)                                    ▷ Next greedy action
10:    $\theta' \leftarrow \theta(s', a')$                                    ▷ Next-state quantiles
11:    target $\leftarrow r + \gamma \cdot \theta'$
12:    Compute pairwise differences $d_{ij} = \text{target}_j - \theta_i$
13:    Compute Huber gradient:

$$g_{ij} = \begin{cases} d_{ij} & |d_{ij}| \leq \kappa \\ \kappa \cdot \text{sign}(d_{ij}) & \text{otherwise} \end{cases}$$

14:    Compute quantile weights:
$$w_{ij} = h_i - \mathbb{I}[d_{ij} < 0]$$

15:    Aggregate gradient:

$$\delta_i = \frac{1}{N} \sum_{j=1}^{N} w_{ij} g_{ij}$$

16:    Update quantiles:
$$\theta_i \leftarrow \theta_i + \alpha \delta_i$$

17: **end function**

---

**Convergence of standard RL models**  To demonstrate that the standard RL models eventually converge to similar reward rates as the TMRL models, we let the models learn for many time steps in the gridworld environment. Figure 13 shows the TMRL model (blue) converges to a similar reward rate as the standard RL model (red). This is using the same learning parameters above with 5 runs of each model.

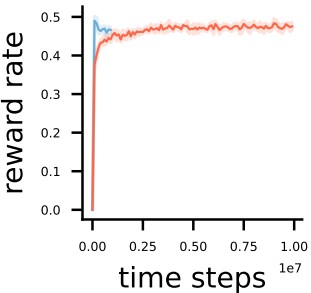 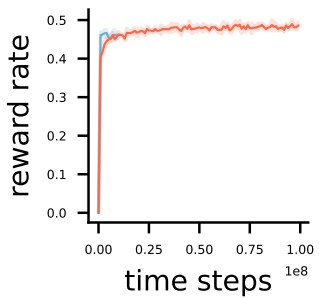 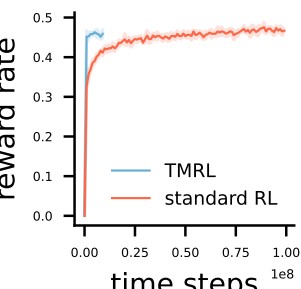

Figure 13: Convergence of standard RL models on the gridworld environment for different number of stimuli. Reward rates here are not normalized. Error bars are standard deviation of 5 runs with different seeds. (a) 3 stimuli (b) 4 stimuli (c) 5 stimuli

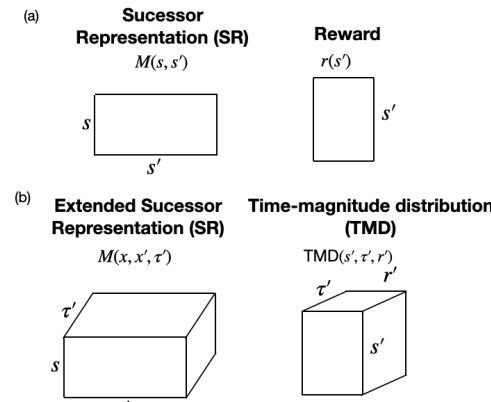

Figure 14: Convergence (a) In order to compute value in a state $s$, the SR over future states is multiplied by the reward vector. (b) We generalize this approach by considering the SR is also defined over future times, so that it may be combined with the TMD representations.

### B.4 MODELING MULTIDIMENSIONAL RISK SENSITIVITY

#### B.4.1 RISK SENSITIVITY IN HUMAN BEHAVIOR

To model the human bandit experiments we considered a set of $N = 10$ units representing the reward time magnitude distribution for each trial type. The subjective value was computed as: $V(s) = \sum_{s',t' \in \{1,...,t_D\},r'} r' \, M(s,s',t')w(t',r')\text{TMD}(s',t',r')$. Importantly, the reward time and magnitudes were normalized by the maximum reward delay and magnitude. This task has a single state, SR is the identity and the weighting for the factorized and multidimensional model are defined in the main text.

#### B.4.2 MAGNITUDE RISK SENSITIVITY

We simulate this agent to display risk sensitivity to uncertainty in reward magnitudes in the gridworld environment. The environment had 3 stimuli: one predicting certain reward of value 2 (certain); another predicting uncertain reward magnitudes of either 1 with 0.5 probability or 7 with 0.5 probability (risky); a control certain reward of value 4 (control). All these rewards had a time delay of 7 time steps after the stimulus. Importantly, the expected value of the risky stimulus is higher than the certain stimulus but is more uncertain. During training of the agents, the stimuli appeared with independent random probability. However, during testing, the certain and risky stimulus were always observed at the same time such that the agent would always have to decide between reward locations.

The magnitude risk sensitive TMRL agent learned the SRs exactly the same as the TMRL agent. However, it learned the TMDs just as matrices. The reward TMD ($TMD_i(s',t_i',r')$ is the probability of reward at location $s'$ of magnitude $r'$ at delay time $t_i'$ after stimulus $i$) was learned by the agent through temporal difference learning Sutton & Barto (2018). For each stimulus $i$ with current delay $t_i^t$, after observing reward of magnitude $r$ at state $s$

$$RMT_i(s',t_i',r') = \begin{cases} 1 & \text{at } s' = s, t_i' = t_i, \text{ and } r' = r \\ 0 & \text{otherwise} \end{cases} \tag{32}$$

$$\delta_{TMD_i}(s',t_i',r') = RMT_i(s',t_i',r') - TMD_i(s',t_i',r') \tag{33}$$

$$TMD_i(s',t_i',r') \leftarrow TMD_i(s',t_i',r') + \alpha\delta_{TMD_i}(s',t_i',r') \tag{34}$$

where $RMT_i(s',t_i',r')$ is just a dummy variable to compute the TD update.

During testing of the magnitude risk sensitive agent, the Q value would be computed by weighting the TMD with weights $w(r)$. These were computed by taking the reward distribution at each reward location at each delay separately and applying a risk function to the distribution over magnitude. This risk function only considers the lower half of the cumulative density function over magnitude.

$$w_{0.5}(r|TMD) = \begin{cases} r & \text{for } r \leq \text{VaR}_{0.5}(r) \\ 0 & \text{otherwise} \end{cases} \tag{35}$$

$$\text{VaR}_{0.5}(r) = \inf\{r \in \mathbb{R} : F_R(r) \geq 0.5\} \tag{36}$$

where $F_R(r)$ is the cumulative density function over magnitudes $r$ of the TMD.

Then the magnitude risk sensitive TMRL agent found the greedy action through GPI while weighting $TMD_{total}^t$

$$Q_j(s_t, a_t) = \sum_r w(r|TMD_{total}^t) TMD_{total}^t(s_{t+1}, 0, r) +$$

$$\sum_{s',t,r'} r' M^{\pi_j}(s_{t+1}, s', t) \cdot w(r|TMD_{total}^t) TMD_{total}^t(s', t+1, r') \tag{37}$$

$$a_t = \text{argmax}_{a_t} \max_j Q_j(s_t, a_t) \tag{38}$$

Supplementary Figure 15 shows the TMRL agent often chose the risky choice with higher expected value but the risk-sensitive TMRL agent almost never chose the risky choice, opting for the certain option. Fig. 15b shows the magnitude risk sensitive agent achieves a slightly lower reward rate compared to the TMRL agent which is directly trying to optimize expected rewards. However, the risk sensitive agent is optimizing to minimize risk.

Parameters for the environment:

- number of runs for each model: 10

- number of states/stimuli: 3

- maximum reward delay: 10

- maximum reward magnitude: 8

- number of training steps: 200000

- number of test steps: 10000

- test every 1000 steps of training

### B.4.3 TIME VARYING RISK SENSITIVITY

We again simulate the agent in the gridworld to show how risk sensitivity can vary through time. Here, the environment had two cues, one with certain reward of value 1 at time delay 7 and the other with certain reward of value 3 at time delay 9. Importantly, during testing, the first stimulus appears with probability 0.2 at every time step while the second stimulus appears with probability 0.1. The expected reward of the second is still larger than the first.

We design an agent that learns the same way as the magnitude risk sensitive agent but the weights depend on an internal state at satiety. The internal state $x_{t+1}$ at time $t+1$ given the reward $R_t$ at time $t$ and the internal state $x_t$ is given by

$$x_{t+1} = \max(2, 0.8x_t + R_t) \tag{39}$$

The internal state decays exponentially over time to zero and increases only when the agent encounters a reward. The agent also has a maximum internal state of 2 signifying the agent is satiated.

We define a subjective value function dependent on $x_t$ as

$$V = (1 - \frac{0.1}{(x_t + 0.1)^2}) \tag{40}$$

This nonlinear subjective value gives higher value to higher internal states $x_t$.

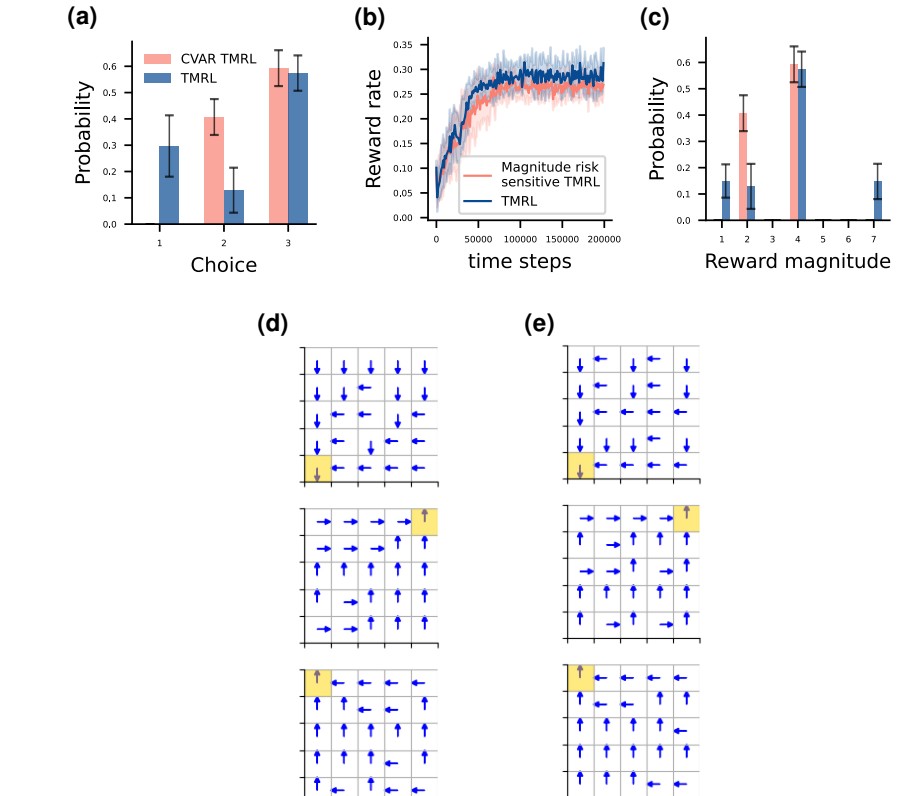

Figure 15: TMRL (blue) and the magnitude risk sensitive agent (pink) in the gridworld environment. (a) The distribution of choices of the agent during testing. (b) The reward rate achieved by the agents through learning. (c) The distribution of reward magnitudes received by the agent during testing. (d) The policy learned by the TMRL agent. Top is the policy when the certain stimulus is observed. The highlighted square is the location of the reward. Middle is the risky reward stimulus. Bottom is the control stimulus. (e) The policy learned by the magnitude risk sensitive agent.

Using this $V$, we compute the weights $w(t, r|x_t)$ by finding the internal states $x_t, ..., x_{t+t_D}$ if a reward of magnitude $r$ were delivered at delay $t$. The weight was then the summed subjective value of these internal states:

$$w(t, r|x_t) = \sum_{t_D \geq t' \geq 0} V(x_{t+t'}) \tag{41}$$

Using these weights we find the GPI action using the same equation 37. Supplementary Fig 16 shows this time-varying risk sensitive agents subjective value compared to that of the TMRL agent. As the time-varying risk sensitive agent is optimizing for this subjective value or utility, it achieves higher values than the TMRL. Yet in Fig 16b, we see the TMRL achieves higher reward rate as it is maximizing expected rewards.

Parameters for the environment:

- number of runs for each model: 10
- number of states/stimuli: 2
- maximum reward delay: 10
- maximum reward magnitude: 4
- number of training steps: 200000
- number of test steps: 10000
- test every 1000 steps of training

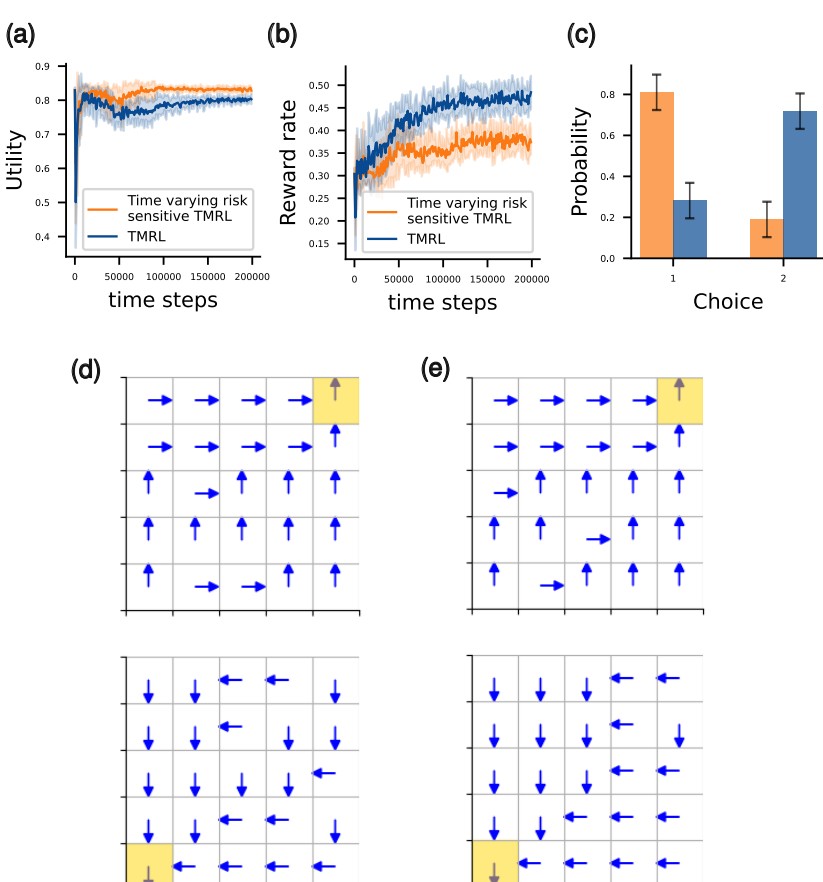

Figure 16: TMRL (blue) and the time-varying risk sensitive agent (orange) in the gridworld environment. (a) The subjective value of the agents over training. (b) The reward rate achieved by the agents through learning. (c) The distribution of choices of the agent during testing. Choice 1 is the high probability, low magnitude stimulus and choice 2 is the low probability, high magnitude stimulus. (d) The policy learned by the TMRL agent. Top is the policy when the low magnitude stimulus is observed. The highlighted square is the location of the reward. Bottom is the high magnitude stimulus. (e) The policy learned by the time-varying risk sensitive agent.

