# OpenReview forum: "Online learning of multidimensional distributional maps for rapid policy adaptation"
_ICLR.cc/2026/Conference — Submitted to ICLR 2026_

### Official Review · Reviewer_yHNF · 2025-10-25

**Soundness:** 3
**Presentation:** 2
**Contribution:** 2
**Rating:** 4
**Confidence:** 2

**Summary:**

This paper lies at the interface of machine learning and neuroscience. The authors present online learning rules for acquiring estimators of multidimensional return distributions in RL, which they posit can be used for efficient risk-sensitive decision-making. They also provide cross-species evidence suggesting that their proposed mechanism is consistent with biological behaviors.

**Strengths:**

* The proposed method is concise and intuitive.
* The authors' claims are well-supported by a comprehensive suite of experiments.

**Weaknesses:**

**Reviewer's Note:** I am a researcher working on reinforcement learning and do not have expertise in brain science. My comments are therefore offered from a machine learning perspective. It is possible that I do not correctly understand some claims in the paper or fully appreciate the significance of the paper's contribution to the field of brain science. Hence, I assign a low confidence score to my evaluation.

* The paper is challenging to follow for readers without specific domain knowledge in neuroscience. To improve accessibility, I strongly recommend adding concise introductions for key concepts (such as efficient coding theory, DAN, and TMD) and including brief overviews of the experimental designs directly in the main text.

* The algorithmic contribution appears limited. In fact, there are already works in distributional RL that address the multi-reward settings like [1], [2], and [3]. The authors should clarify how their approach differs from or improves upon these existing works.

* The related work section currently overlooks relevant bodies of work on generative models (e.g., VAEs, GANs, diffusion models) and other recent advances in distributional RL. I encourage the authors to conduct a more thorough review to better position their contributions within these contexts.

[1] Distributional Reinforcement Learning with Regularized Wasserstein Loss
[2] Distributional Multivariate Policy Evaluation and Exploration with the Bellman GAN
[3] GAN Q-learning

**Questions:**

The authors combine a KL objective with Wasserstein proximity to update distribution representations. Could the authors please clarify the main advantages of this specific technique? A brief discussion justifying this choice over other distributional update rules would be beneficial.

---

> ### Author Response · Authors · 2025-11-25
>
> **Weakness 1**
>
> 1) Thank you for the recommendations, we have now used the extra space to include a figure describing efficient coding theory (Updated Fig. 1a). We also now introduce efficient coding theory in the introduction in a clearer and more accessible way for readers who may be unfamiliar with the concept (line 057):
>
> *In neuroscience, the theory of efficient coding has modeled how sets or populations of neurons maximize the amount of information they encode about diverse stimuli in the environment.*
>
> Finally, we have added a section (Section 2) where we describe in technical terms neuroscience efficient coding theory.
>
> 2) We are now defining cleary TMD in line 213:
>
> *...we focus on the TMD, the joint distribution of reward times ($\tau$) and magnitudes ($m$): $p(\tau, m)$...*
>
>
> **Weakness 2 and Question**
>
> In the related work section we compare the difference between our approach and the approaches in refs [1,2], which we have now updates to also contain ref 3:
>
> *In machine learning, mean maximum mean discrepancy (MMD) and regularized Wasserstein losses have been applied to learn multidimensional distributions (Zhang et al 2021, Sun et al. 2024, Wiltzer et al 2024). However, these approaches regularize the optimized distribution, whereas DNL regularizes the learning trajectory. In dimensions bigger than one, this distinction is critical: there are many distinct particle configurations that represent the same distribution, which leads to degenerate solutions (Fig. 1).
> We address this degeneracy by introducing a Wasserstein regularizer on the smoothness of the learning trajectory, enforcing the conservation of relative particle positions throughout learning. This modification yields three key benefits:
> (1) it extends efficient coding principles to higher dimensions (Schutt et al 2024);
> (2) it preserves relative particle arrangements as the reward distribution changes, matching empirical observations across multiple brain areas and modalities; and
> (3) it supports flexible decoding of subjective value (i.e., risk-sensitive utility) under context-dependent time–magnitude distributions (TMDs), removing the need to retrain the decoder.*
>
> The fundamental qualitative distinction is that we are regularizing the learning trajectory, while these references are regularizing the optimized distribution. We thank the reviewer for pointing out these relevant references. Reference [1] has been added.
>
> **Weakness 3**
>
> In order to broaden our literature review and to better contextualize our work, we will add an extended “related work” section in the SM to better position our work relative to other work in generative modeling and distributional RL.

---

### Official Review · Reviewer_B1a5 · 2025-10-26

**Soundness:** 2
**Presentation:** 1
**Contribution:** 1
**Rating:** 2
**Confidence:** 3

**Summary:**

Inspired by neuroscience, this paper proposes the online updating rule of the time-magnitude distributions (TMD) by using multi-dimensional distributional RL. The proposed algorithm enables rapid risk-sensitive action selection in environments. This work further connects distributional RL and neural coding and experiments are performed on neuroscience and gridworld environments.

**Strengths:**

1.This paper further strengthens the connection between distributional RL and neuroscience, which provides a new computational perspective in this area.

2.The illustration looks impressive, with many new environments from the neuroscience.

**Weaknesses:**

1. **Limited Motivation and Technical Contribution**. This paper seems to be an application of the multi-dimensional distributional RL algorithm to develop the online update rule of time-magnitude distributions (TMD) and the modifications are straightforward. From the perspective of (distributional) RL, this paper may not be well-suited to ICLR, as the focus is mainly on the neurocoding side. In particular, there are already many multi-dimensional distributional RL algorithms, such as [1, 2, 3], that present competitive performance in deep RL settings. Thus, it is less motivated to posit the contribution of this paper in the distributional RL literature from the perspective of technical contribution. It may be a useful extension in the neurocoding theory, which is beyond my expertise.

2. **Unclear writing**. It is less clear to me about many technical words and descriptions in the context of neuroscience and neurocoding theory. For instance, what is the time-magnitude distribution in a mathematical way? Why do the authors claim that distributional RL is largely used for improvement for improving representation learning in Line 22? Any reference about that? There are also a lot of risk-sensitive RL papers that are related to this work, which has not been sufficiently investigated here.

3. **Limited experiments and insufficient explanations**. The references on the figures are provided without detailed explanations. Thus, it is easily confused about the connection between the experimental results and the claim the authors want to make. Some explanations are only given in the captions, but detailed elaborations are more important in the main content. In addition, the experiments are only performed on some neuroscience and gridworld environments in the tabular setting, which is far beyond the conventional environmental setup. Large-scale environments are almost necessary, especially in RL literature.


## Reference
[1] Zhang, Pushi, et al. "Distributional reinforcement learning for multi-dimensional reward functions." Advances in Neural Information Processing Systems 34 (2021): 1519-1529.

[2] Sun, Ke, et al. "Distributional reinforcement learning with regularized wasserstein loss." Advances in Neural Information Processing Systems 37 (2024): 63184-63221.

[3] Wiltzer, Harley, et al. "Foundations of multivariate distributional reinforcement learning." Advances in Neural Information Processing Systems 37 (2024): 101297-101336.

**Questions:**

Beyond the question in the weaknesses, here are some other questions.

1. What are the benefits of adding a Wasserstein regularizer? How does it address the limitations of Zhang et al 2021 mentioned in Line 102? In my opinion, the motivation is not clear.

2. It is not clear why the authors introduce the definitions in Eq.3. The authors mentioned an entropic regularization in Eq. 5. What is the overall objective function, and is there any relationship or difference between the Sinkhorn divergence proposed by [2]?

3. The authors mentioned SM many times in the paper, but there is no explicit reference on the section, making it unclear about the detailed section in the appendix.

4. The explanation in Section 2.3 is described in a sloppy way, which is hard to follow.

5. The experimental details in the context of neuroscience should be elaborated further. For example, the patch environment used here is not standard in RL literature. For the current version, the choice of environment is too specific, which is tailored for neuroscience readers.

6. Why not follow the experimental setup in Implicit Q-learning to evaluate the risk-sensitive behavior in Section 5?

---

> ### Author Response · Authors · 2025-11-25
>
> **Questions**
> 1. In this work we unified the neuroscience theory of efficient coding with distributional reinforcement learning. Signatures of efficient coding have been found in many areas of the brain, and the brain encodes multidimensional features of the environment, however these models don’t generalize to multi-dimensional distributions, and online-learning rules are lacking. In order to generalize efficient codes to dimensions higher than one, it is necessary to add the Wasserstein regularizer in the learning trajectory. The references [1, 2, 3] are not directly related to efficient neural population coding. These works focus on regularizing the target distribution, whereas DNL regularizes the learning trajectory itself. Adding this regularizer, enforces the conservation of relative positions of the particles and the learning trajectories are smoother (Figure 1c), which also leads to a faster adaptation of linear risk-sensitive value decoders (Figure 2c bottom right), compared to refs [1,3]. We have substantially revised the Related Work section to provide a clearer and more coherent explanation of the underlying motivation.
> 2. The two main differences with the Sinkhorn divergence proposed in [2] are: 1) in this work the value distribution is learnt through distributional Bellman updates, whereas we are learning the reward distributions; 2) we are minimizing the Wasserstein across different iteration distributions, whereas ref. 2 is minimizing the Wasserstein between the Bellman target and the current value distribution. Put more simply, the fundamental qualitative distinction is that we are regularizing the learning trajectory, while these references are regularizing the optimized distribution. Crucially, we are specifically optimizing Eq. 3 (updated Eq. 4) because it generalizes the efficient coding theory to dimensions bigger than one. We have now added a more detailed explanation of why we are optimizing updated Eq. 4 in the Section 2.1.
>
> 3. Thank you for the note, we will specify the SM section.
> 4. Thank you for the note. We have devoted more space for the derivations and believe now it is clearer. The full derivation is in Section A.4.
> 5. We will clarify.
> 6. The implicit Q-learning approach does not incorporate risk modulated in time.
>
> **Others:**
>
> The time-magnitude distribution (TMD) is the joint distribution of reward times ($\tau$) and magnitudes ($m$): $p(\tau,m)= \frac{1}{N} \sum_{n=1}^N \delta(\tau - \tau_n, m - m_n)$, that is approximated non-parametrically using delta functions.
>
> References where it was shown that distributional RL specifically improves representation learning:
>
> Marc G Bellemare, Will Dabney, and Rémi Munos. A distributional perspective on reinforcement learning. In International conference on machine learning, pp. 449–458. PMLR, 2017
>
> Will Dabney, Mark Rowland, Marc Bellemare, and Rémi Munos. Distributional reinforcement
> learning with quantile regression. In Proceedings of the AAAI Conference on Artificial Intelligence, volume 32 (1), 2018b.
>
> Mark Rowland, Robert Dadashi, Saurabh Kumar, Rémi Munos, Marc G Bellemare, and Will Dabney. Statistics and samples in distributional reinforcement learning. In International Conference on Machine Learning, pp. 5528–5536. PMLR, 2019
>
> Thank you for pointing out that this line of work should not be assumed to be common knowledge and now reference appropriately in our paper.
>
> We have cited previous work on risk-sensitive RL, e.g. Dabney et al. (2018a) and Ávila Pires et al. (2025), please share with us the specific references you think we are missing. The main advantages of our method are: 1)  it can adapt fastly to changes in the reward time magnitude distributions (Fig. 2c); 2) it allows for generating risk sensitive behavior that varies in time (Fig. 5).

---

### Official Review · Reviewer_PBjw · 2025-10-28

**Soundness:** 3
**Presentation:** 2
**Contribution:** 3
**Rating:** 4
**Confidence:** 2

**Summary:**

This paper focuses on an empirical finding of previous study (Sousa et al.,2025), that the midbrain dopamine neurons in the brain encode the time-magnitude distributions (TMD) of (multi-dimensional) rewards.
The core goal of this paper is to learn the TMD distributions efficiently, the paper proposes an online method that minimizes the Kullback-Leibler (KL) divergence with W_2 distance regularization.
Through extensive experiments, the paper demonstrates that the proposed method outperforms solutions based on traditional distributional reinforcement learning modeling.

**Strengths:**

In my view, the problem considered in this paper is valuable: how to make decisions when rewards can only be observed after a certain delay.
The paper provides a model and an algorithm for this problem and conducts experiments across several well-designed experimental environments.
The experimental results demonstrate the effectiveness of their method compared to traditional approaches including quantile-based distributional RL algorithms.

**Weaknesses:**

As a reviewer without expertise in neuroscience, I find it difficult to accurately assess the contributions of this paper. From the perspective of machine learning alone, the approach and analysis proposed in Section 2.1, using W_2 regularization to prevent collapse and maintain diversity, is a common practice. It is also predictable that this regularization would lead to smoother solutions.

I recommend that the authors consider submitting this work to more specialized neuroscience journals (e.g., Nature or its sub-journals) to obtain more accurate evaluations .

**Questions:**

In some experiments, such as those presented in Figure 3, the setup appears to violate the Markov property required for RL algorithms. Given this, it seems that RL algorithms—whether distributional or traditional methods that only model value functions—should be unable to learn optimal policies. Is my understanding correct?

---

> ### Author Response · Authors · 2025-11-25
>
> **Questions:**
> Thank you for the question. The Markov property would be violated without the correct state space. For the standard RL agents, we must use an expanded state space specifically to avoid violating the Markov property. This expanded state space gives a unique state for every combination of location and time since each stimulus as described in line 315 for the patch experiment and line 334 for the gridworld. This means the standard RL agents should eventually learn the optimal policy. We will include a statement in the appendix B.3.1 to address this question:
>
> *The state space is constructed this way to ensure that the Markov property is satisfied.*
>
> We also include supplementary figure 13 with final converged performance of all the standard models to show they can reach the same performance as the TMRL models. They just take many more steps in the environment to converge.
>
> **Weaknesses:**
>
> 1. We believe ICLR is an appropriate venue for our work as “applications to neuroscience” are described as a relevant topic for ICLR. Indeed, we have marked the primary area for this submission as applications to neuroscience & cognitive science.
>
> 2. In this work we generalize efficient neural population coding to multiple dimensions and provide learning rules, which has not been done before. In doing so, we establish a novel link between theories of cortical coding and distributional RL.
>
> 3. Indeed, previous work has used Wasserstein regularization. A key distinction with respect to our work is that we employ Wasserstein regularization across successive iterations during the learning process rather than Wasserstein-regularization of the optimized distribution. We understand that the technical motivation for doing this was not clear enough and therefore we have now added a more detailed explanation of why we regularize via the Wasserstein distance across successive iterations distributions.
> 4. The core technical idea is that evolving an initial probability distribution toward a target distribution can be described by the Fokker–Planck equation (FPE). Specifically, an efficient neural code can be defined as a solution of a FPE. Importantly, a classical result from fluid dynamics (Jordan, Kinderlehrer, & Otto, 1998) shows that the solution to the FPE corresponds to a Wasserstein gradient flow on a free‐energy functional, such as the Kullback–Leibler (KL) divergence, in the space of probability measures, which provides a formalism for online optimization. We adapt this formalism to provide learning rules that correspond to distributional RL update rules in a particular limit. Thus, a consideration of Wasserstein regularization on iterative distributional updates is necessary for our theory. The revised Section 2.1 now provides the relevant technical details that resolve this issue.

---

### Official Review · Reviewer_vCKN · 2025-10-30

**Soundness:** 3
**Presentation:** 1
**Contribution:** 2
**Rating:** 6
**Confidence:** 3

**Summary:**

This paper considers the problem of learning and using multi-dimensional reward distributions. It asks how to represent such distributions, how to learn the representations, and how to use these representations for action selection. To address these questions, the paper brings together approaches from three related disciplines: distributional RL, efficient coding, and risk-aware decision making. The authors derive a learning rule that encourages proximity to a target distribution but also efficient trajectories of approaching the target, which improves multi-dimensional distribution learning. They then show that this rule is consistent with empirical dopamine neurons, that agents that learn time-magnitude distributions outperform agents that don’t in tasks with particular reward structure, and how knowing the time-magnitude reward distribution can support risk-aware decisions.

**Strengths:**

This has the potential to be a good paper. It’s an interesting question, a nice approach, and the results are generally solid. The learning trajectory regularization is clever. It’s great that there’s a comparison to empirical dopamine neuron activity. The results on agents that learn time-magnitude distributions are convincing (although by design: learning a full enumerated state space must be slower than one that has part of the true reward structure baked in). The asymmetric discounting that only works for multidimensional and not for factorised discounting is a neat insight; I’m not too familiar with this literature so I’m not completely sure it’s novel, but it fits well within the paper’s story regardless. Overall, I think the experiments and results are useful though not groundbreaking. I'm leaning to accept.

**Weaknesses:**

As mentioned above, I think the controls are on the weak side. While this is useful to illustrate qualitatively what the new methods brings, it makes it hard to estimate the impact when evaluated against less limited models. Specifically, this applies to the factorised distribution in Fig 1b, the lack of control methods in Fig 2c, the standard/QR RL models in Fig 3b,e and the non-converged agents in Fig 3c,f. Additionally, it would be in the authors’ best interest to thoroughly proofread the manuscript before submission, as in its current form it is unnecessarily difficult to read, which risks alienating both reviewers and readers. In particular the repeated mislabelling of figures caused confusion.

**Questions:**

Major comments

The various writing mistakes, from grammar and typos to repeated figure panel mislabeling, contributed to making this paper difficult to read at times. I’ve included examples of such cases from just the first two pages below this review, but across the document it’s hard not to get a feeling of a lack of attention to (and proofreading of) the writing. There are multiple instances where a careful check of the text and figures could have avoided confusion. Figure panels are not referenced (e.g. Fig 1b is never mentioned in the text) or don’t exist (e.g. Fig 1d is referenced but there are only three panels in Fig 1; Fig 2d is referenced but again doesn’t exist; Fig 4d-f referenced but don’t exist (line 424); Fig 4g-l referenced but don’t exist (line 461)) or are mislabeled (e.g. Fig 1c in the main text seems to refer to the result of Fig 1a; Fig 4b in line 416 seems to refer to Fig 4c; Figure 4b in line 424 doesn’t seem to refer to Fig 4b). The fact that the references are very messy, with many items occurring multiple times in slightly different formats, doesn’t help to convey an impression of attention to detail either.

Figure 1a is referenced once, directly before Equation 4, but the result in the panel seems unrelated to Equation 4. It took me a while to understand what was plotted in Figure 1a because the caption is very minimal, and the main text also didn’t explain the result of that panel. Then I got to section 2.3 which seems to exactly explain Figure 1a, but it refers to Figure 1c. Moreover, this section 2.3 comes after section 2.2 which explains the results of Figure 1b and 1c. The fact that 1) the reference to figure panels in the main text doesn’t match the actual figure panels and 2) the results in the main text are in a different order from the figure panels made this confusing.

I don’t think the factorized 1D distribution learning is a particularly strong control in Figure 1b. It seems obvious that a 1D distribution learning method won’t be able to learn correlations between variables. Maybe it would be useful to add another multi-dimensional method, like MMD, as a panel? That would potentially make the point that while MMD can learn the same distribution as DNL (which this new panel would show), its learning trajectory is different (as shown in current panel 1c).

Does the gamma before vs gamma after result in Fig 2c only appear in DNL? Would other distribution learning methods, or even a factorized 1d distribution learning method, not produce the same result?

The results in Fig 3c,f are superfluous because they can be directly read from 3b,e by only looking at the final datapoint. It feels a bit unfair to only compare performance for a fixed number of timesteps – it is expected that an agent with access to a usefully structured state space learns faster than an agent that must enumerate all states, and this difference is already shown in 3b,e. The fairer comparison in 3c,f would be to compare performance after convergence of all agents.

Minor comments

147 What is \Delta_i ?

042 Unclear what TMD abbreviates as the preceding words don’t match the letters “extend the 1D reward magnitude code to a 2D time-magnitude “map” of future reward in a  distributional format (TMD)”.

047 “In particular, in the naturalistic scenario of choosing between actions leading to probabilistic rewards generated by the environments with intricate temporal structure and distributional shifts.” > Missing a verb.

098 “proposed maximum mean discrepancy (MMD) based multidimensional distributional algorithm similar to DNL” > missing “a” maximum (…) algorithm.

099 DNL acronym used without prior introduction.

102 “We address this by adding a Wasserstein regularizer that penalizing distortions in the population representation” > that “penalises”.

104 “which has three important consequences (1) extends efficient coding to higher dimensions (Schütt et al., 2024), (2) preserves the population coding when the reward distribution changes as observed in many brain regions across multiple modalities and (3) enables flexible decoding” > missing “:” after consequences, and “it” before (1) extends

211 Absorves > absorbs

456 (c) > (e)

Fig 4b No legend for blue bar (I guess it corresponds to TMRL in panel a but would be useful to be explicit)

---

> ### Author Response · Authors · 2025-11-25
>
> Thank you for pointing out the typos, we have corrected them.
>
> **Question: I don’t think the factorized 1D distribution learning is a particularly strong control in Figure 1b.**
>
> Thank you for pointing this out. We are adding the MMD control to panel b.
>
> **Question: Does the gamma before vs gamma after result in Fig 2c only appear in DNL? Would other distribution learning methods, or even a factorized 1d distribution learning method, not produce the same result?**
>
> Yes, it would. However, for more complex high dimensional distributions, the MMD does not preserve population geometry and this has two main problems:
>
> 1) There is evidence that the tuning of midbrain dopamine neurons is organized topographically, and that the mechanism used to decode reward information and select actions relies on this topographic organization (Refs 1-4). If across different tasks, the distribution of reward time-magnitude distributions shifts, the decoding of risk sensitive value for MMD does not adapt immediately, but for DNL does (Fig. 2c).
>
> Ref 1: Watabe-Uchida, M., Zhu, L., Ogawa, S.K., Vamanrao, A. and Uchida, N., 2012. Whole-brain mapping of direct inputs to midbrain dopamine neurons. Neuron, 74(5), pp.858-873.
>
> Ref 2: Hunnicutt, B.J., Jongbloets, B.C., Birdsong, W.T., Gertz, K.J., Zhong, H. and Mao, T., 2016. A comprehensive excitatory input map of the striatum reveals novel functional organization. elife, 5, p.e19103.
>
> Ref 3: Foster, N.N., Barry, J., Korobkova, L., Garcia, L., Gao, L., Becerra, M., Sherafat, Y., Peng, B., Li, X., Choi, J.H. and Gou, L., 2021. The mouse cortico–basal ganglia–thalamic network. Nature, 598(7879), pp.188-194.
>
> Ref 4: Björklund, A. and Dunnett, S.B., 2007. Dopamine neuron systems in the brain: an update. Trends in neurosciences, 30(5), pp.194-202.
>
> 2) In the SM section A.2 and Fig. 7 an example that disambiguates DNL and MMD is given. It is a spatial cognitive task, as described in Krupic et al 2015, where the environment shape changes: it initially is a rectangle and then the lower triangular section of the environment is removed. In this case DNL accurately models the place cell adaptation, however the MMD does not (Compare SM Fig. 7a,b). We would be happy to move this example to the main paper.
>
>
> **Question: The fairer comparison in 3c,f would be to compare performance after convergence of all agents.**
>
> The final performance of the TMRL agents after convergence is actually equal to the standard RL agents. This is because the standard RL agents with the enumerated state space actually makes this a markovian state space. So the standard agents should also be able to learn to optimality. We simply want to highlight the learning speed, an important aspect for an animal agent. We include a supplementary figure 13 with final converged performance of all the standard models.
>
> **Minor**
>
> $\Delta_i$ is an update on $\theta_i$.
>
> We thank the reviewer for this detailed feedback and apologize for the unnecessary grammatical errors. These have all been resolved in the updated version of our paper.
>
> Section 2 has been rewritten to improve clarity. It now presents the derivation of the DNL update rules in multiple subsections and includes a new figure illustrating the main theoretical components: efficient coding, distributional RL, and optimal transport.

---

> > ### Comment · Area_Chair_RoNv · 2025-11-26
> >
> > Dear Authors,
> >
> > Are you planning to upload a revised PDF? This would help the review team ensure that the concerns have been fully addressed.
> >
> > Thanks!
> >
> > --AC

---

> > > ### Author Response · Authors · 2025-11-28
> > >
> > > Dear AC,
> > >
> > > We apologize for the delay, we were finalizing an additional Supplementary Figure. We have now uploaded a revised pdf.
> > >
> > > Thank you!

---

### Official Review · Reviewer_rDPg · 2025-11-01

**Soundness:** 3
**Presentation:** 1
**Contribution:** 2
**Rating:** 0
**Confidence:** 3

**Summary:**

Contemporary RL work mostly focus on discounted future return and there has been lack of studies focusing on the impact of reward timing and reward magnitude. In this paper, the authors addressed these limitations by introducing multi-dimensional distributional neural learning (DNL) which is based on optimal transport theory.  According to the authors, the parameters of the tuning functions are learned using DNL, which the results also align with empirical observations relating to distributions of reward times and magnitudes. After the parameters of the tuning functions have been learned, the time-magnitude distributions (TMD) can then be approximated and actions are selected based in the TMD. The authors also presented a grid world setup where the TMD can be combined with Successor Representations to learn the value function, resulting in the TMRL agent. Compared to the standard Q-learning agent and the distributional RL agent, the TMRL manage to learn more effectively, even as the number of stimuli increases. Furthermore, the authors also extend the studies to include certain and risky rewards and showed that the Multidimensional representations yield better results.

**Strengths:**

1.  Idea of the paper involves Reinforcement learning and provide insights into representations that align well with empirical data.
2. Most of the important equations are provided in the main paper.

**Weaknesses:**

1. The paper is poorly written. There are a lot of complex concepts in the paper that were not introduced gracefully. Furthermore, some concepts were not explained why they are required. For example, why was Successor Representations required for the gridworld experiment? Why was the w variable required?

2. For readers who are not familiar with prior work, reading this paper is very challenging. The flow between some of the paragraphs were not very fluid. For example, the w variable was not defined when first introduced at line 361.

3. Some of the figures caption are labeled incorrectly or mentioned in the main text but the figure is non-existent. For example, where is Figure 1d (line 220)? Figure 4e has been mistakenly been labeled as Figure 4c in the caption. Line 345 mentioned Figure 3d as a results but Figure 3d is a figure of the grid world environment.

4. With the above reasons and the current state of the paper, at this moment, I feel that the paper is rushed and not ready for publication.

**Questions:**

1. What is DNL? (Line 99). Looks like it is "Distributional neural learning". The abbreviation should be introduced much earlier, rather than at line 203.
2. The last caption in Figure 4 should be (e) instead of (c).
3. What is the dimension of the output of the TMD function? It is not stated clearly.
4. It is unclear how action selection is done for the TMRL agent. After some digging, the information can be found in the appendix but the authors should have mentioned about the existence of the pseudocode in the main paper.
5. I also listed some questions / remarks in the weakness section.

---

> ### Author Response · Authors · 2025-11-25
>
> **Strengths**
>
> We believe that there are more specific and meaningful strengths of our study.
>
> **Weakness**
> 1. Our justification for the use of successor representations starts at line 398:
>
> *By combining these SRs with the current probabilistic reward time-magnitude
> distribution TMD(s′, τ ′, r) determined by the timings of recent stimuli, the agent is able to use
> generalized policy improvement (GPI) to select actions.*
>
> We will now include a statement when introducing the SR:
>
> *However, the TMRL agent now learns multiple successor representations (SR) (Dayan, 1993) expanded in delay time as well as a TMD for each of the stimuli, as depicted in Updated Supp. Fig. 12. The successor representations learn separate policies that can bring the agent to different reward locations and allow flexible combinations of these policies using generalized policy improvement (GPI) when multiple rewards are available with various time delays.*
>
> 2. Due to the page limit we had informally defined the weighting function:
>
> *For 1-dimensional reward magnitude distributions, risk sensitivity behavior can be generated by assigning weights to the reward distribution quantiles (Dabney et al., 2018a). For example, overweighting lower reward quantiles generates risk-averse behavior, while overweighting higher quantiles instead produces risk-prone behavior. We extend risk sensitivity to reward time and magnitude by applying weights that depend on both reward magnitudes and delays to compute the subjective value: ….. To model risk sensitivity in sequential RL tasks, we consider the gridworld environment in Updated Fig. 3d.*
>
>  We have now formally defined the weighting function:
>
> *To model risk sensitivity in sequential RL tasks, we consider the gridworld environment in Updated Fig. 4d with different stimuli structure and weighting functions $w: \mathbb{R}^2 \rightarrow \mathbb{R}$, such that $w(\tau, r)$ is the weight assigned to the reward time $\tau$ and magnitude $r$.*
>
> 3. Thank you for catching these. We have solved these typos.
>
> **Questions**
>
> 1. Thank you for pointing this out. We have now included the acronym earlier in the abstract in line 018 :
>
> *Here we present distributional neural learning (DNL), online learning rules for acquiring information-maximising multidimensional distributional estimates, extending classic work in distributional RL from 1D return distributions to efficient representations of distributions of arbitrary dimensionality.*
>
> 2. Thank you again for noticing this typo.
>
> 3. We are now clearly defining it in end of updated section 2.1:
>
> *...we focus on the TMD, the joint distribution of reward times ($\tau$) and magnitudes ($m$): $p(\tau, m)$...*
>
> 4. In the main paper, we now direct the reader to the algorithm in the appendix:
>
> *Further details of the agent implementations including learning of the SR and action selection can be found in Algorithm 1, 2, 3, and 4 of the Appendix.*
>
> Section 2 has been rewritten for clarity using the extra space afforded in the extended format. It now presents the derivation of the DNL update rules in multiple subsections and includes a new figure illustrating the main theoretical components: efficient coding, distributional RL, and optimal transport.

---

> > ### Comment · Reviewer_rDPg · 2025-11-27
> >
> > Thank you addressing some of the concerns. I have the following point while waiting for the pdf to be updated.
> >
> > 1. I think the flow of the readability can be addressed by a preliminary section, where the important concepts, such as RL, SR are introduced. I apologies for not mentioning this early so I won't penalise authors on this.

---

> > > ### Author Response · Authors · 2025-11-28
> > >
> > > Thank you for the time.
> > >
> > > 1) We have added a sentence in the Introduction describing RL in general terms:
> > >
> > > *One of the most fruitful intersections between natural and artificial intelligence research has been the
> > > idea that midbrain dopamine neurons (DANs) in the brain encode a reward prediction error critical
> > > for reinforcement learning (RL) (Schultz et al., 1997), a suite of algorithms enabling an agent to learn
> > > to select actions based on reward feedback (Sutton & Barto, 2018).*
> > >
> > > 2) For a definition of SR we direct the reviewer to lines 389-400 in the main text and to section B.2.3 lines 1405-1457 of the appendix.

---

### Meta-Review · Area_Chair_Bj9f · 2026-01-07

**Summary:**

This paper presents a computational model for neuroscience that disentangles the timing and magnitude of the reward when estimating temporally discounted average future reward. It does so through multidimensional distributional estimates. Among other things, it also introduces a control algorithm that leverages such estimates.

Reviewers had two main concerns: the paper's presentation and its fit with ICLR. In fact, the paper suffered from receiving reviews from people mostly outside of neuroscience. In any case, while I do think the topic is of interest to the ICLR community, the paper does have enough presentation issues that prevent its publication, and I am recommending its rejection. To be specific, I am not talking about the difficult reviewers outside the field had to understand the paper, but about the fact that the paper flow is challenging to follow, and that it has many typos, grammar errors, references in different formats, mislabeled figures, and more. Some of that was addressed in the new version, but given the shortened review period and the fact that reviewers were cut out of the loop, this paper simply doesn't seem ready for publication yet.

**Reviewer Concerns:**

- Presentation: the paper is hard to read both because of its flow and because of its many typos, grammar errors, references in different formats, mislabeled figures.
- Fit to conference: the paper is a computational model for neuroscience, something many ICLR reviewers are not familiar with.

**Reviewer Scores:**

Reviewer rDPg -> Initial rating was 0 with a review that was not so informative. I don't know what the reivewer would do, but I don't think we should take the 0 into consideration here.
Reviewer vCKN -> Initial rating was 6, would likely keep the same rating.
Reviewer PBjw -> Initial rating was 4, would likely keep the same rating.
Reviewer B1a5 -> Initial rating was 2, would likely keep the same rating.
Reviewer yHNF -> Initial rating was 4, would likely keep the same rating.

---

### Decision · Program_Chairs · 2026-01-26

Reject